# Teacher's pet: understanding and mitigating biases in distillation

**Michal Lukasik**  *mlukasik@google.com*
*Google Research*

**Srinadh Bhojanapalli**  *bsrinadh@google.com*
*Google Research*

**Aditya Krishna Menon**  *adityakmenon@google.com*
*Google Research*

**Sanjiv Kumar**  *sanjivk@google.com*
*Google Research*

**Reviewed on OpenReview:** *https://openreview.net/forum?id=ph3AYXpwEb*

## Abstract

Knowledge distillation is widely used as a means of improving the performance of a relatively simple "student" model using the predictions from a complex "teacher" model. Several works have shown that distillation significantly boosts the student's *overall* performance; however, are these gains uniform across all data subgroups? In this paper, we show that distillation can *harm* performance on certain subgroups, e.g., classes with few associated samples, compared to the vanilla student trained using the one-hot labels. We trace this behaviour to errors made by the teacher distribution being transferred to and *amplified* by the student model, and formally prove that distillation can indeed harm underrepresented subgroups in certain regression settings. To mitigate this problem, we present techniques which soften the teacher influence for subgroups where it is less reliable. Experiments on several image classification benchmarks show that these modifications of distillation maintain boost in overall accuracy, while additionally ensuring improvement in subgroup performance.

## 1 Introduction

Knowledge distillation is a technique for improving the performance of a "student" model using the predictions from a "teacher" model. At its core, distillation involves replacing the one-hot training labels with the teacher's predicted label distribution. Empirically, distillation has proven successful for model compression (Bucilă et al., 2006; Hinton et al., 2015), improving the performance of a fixed model architecture (Anil et al., 2018; Furlanello et al., 2018), and semi-supervised learning (Radosavovic et al., 2018). Theoretically, several works (Lopez-Paz et al., 2016; Mobahi et al., 2020; Tang et al., 2020; Menon et al., 2020; Zhang & Sabuncu, 2020; Ji & Zhu, 2020; Allen-Zhu & Li, 2020; Zhou et al., 2021; Dao et al., 2021) have studied how distillation affects learning. Both strands of work further the understanding of when and why distillation helps.

In this paper, we are similarly motivated to better understand the mechanics of distillation, but pose a slightly different question: does distillation help *all* data subgroups uniformly? Or, do its overall gains come at the expense of *degradation* of performance on certain subgroups? To our knowledge, there has been no systematic study (empirical or otherwise) of this question. This consideration is topical given the study of *fairness* of machine learning algorithms on under-represented subgroups (Hardt et al., 2016; Buolamwini & Gebru, 2018; Chzhen et al., 2019), and the study of the tension between *average* and *subgroup* performance of common learning algorithms (Samadi et al., 2018; Sagawa et al., 2020a; Jones et al., 2021).

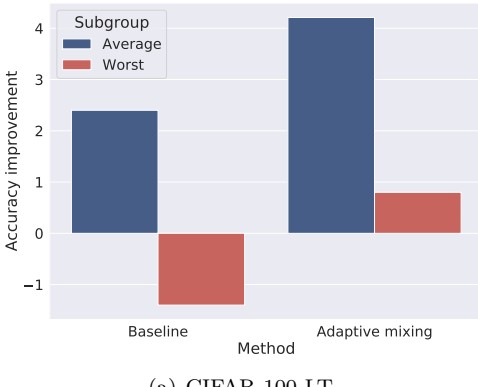

(a) CIFAR-100-LT.

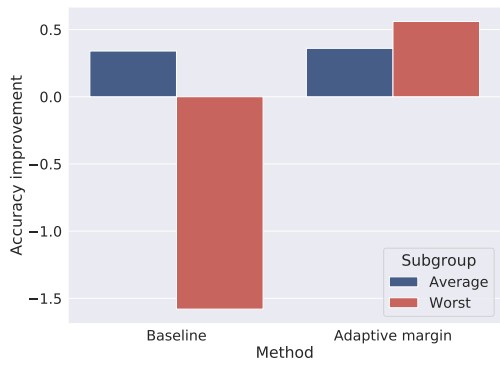

(b) ImageNet.

Figure 1: Illustration of the potential deleterious effects of distillation on data subgroups. We train a ResNet-56 teacher on **CIFAR-100-LT**, a long-tailed version of CIFAR-100 (Cui et al., 2019; Cao et al., 2019) where some labels have only a few associated samples, and a ResNet-50 teacher on **ImageNet**. For each dataset, we self-distill to a student ResNet of the same depth. On CIFAR-100-LT, as is often observed, distillation helps the *overall* accuracy over one-hot student training ($\sim$2% absolute). However, such gains come at significant cost on subgroups defined by the individual classes: on the ten rarest classes, distillation *harms* performance by $\sim$1%. Similarly, on ImageNet, distillation harms the average accuracy of the worst-10 classes (as determined by the teacher's performance) by $\sim$1.5%. Our proposed techniques (§4) can roughly preserve the overall accuracy, while boosting subgroup performance. Notice that the label frequency may not be always consistent with performance or performance improvement in an unbalanced dataset. However, we find them to be strongly related: for CIFAR-100 LT, Spearman rank correlation coefficient between the ordering according to frequencies and according to teacher's performance is equal to 0.84.

Our first finding is that even in standard settings — e.g., on image classification benchmarks such as CIFAR — distillation can *disproportionately harm* performance on subgroups defined by the individual classes (see Figure 1). Specifically, compared to the teacher model, distillation can *worsen* the performance on hard classes, and *amplify* the gap between worst- and average-class performance. To discern the source of this behaviour, we ablate the teacher and student architectures (§3.2), dataset complexity (§3.3), label frequencies (§3.4). These point to the potential harms of distillation when the teacher *confidently mispredicts* on a subgroup.

Having identified a potential limitation of distillation, we present two simple techniques to remedy it. These apply per-subgroup mixing weights between the teacher and one-hot labels, and per-subgroup margins respectively (§4). Intuitively, these limit the influence of teacher predictions on subgroups it models poorly. Experiments on image classification benchmarks show that these methods typically maintain a boost in overall accuracy, while ensuring a more equitable improvement across subgroups.

In sum, this work provides novel insights into distillation performance, with the following contributions:

(i) we identify a hitherto unexplored issue with distillation, namely, that its improvements in overall accuracy may come at the expense of harming accuracy on certain subgroups (§3.1). Such a finding is topical given the widespread practical use of generic learning paradigms such as distillation, and the increasing societal applications of learning systems more broadly.

(ii) we ablate potential sources for the above phenomenon (§3.2, §3.3, §3.4), and in the process identify certain characteristics of data (e.g., skewed label distributions) where it can manifest.

(iii) we provide a theoretical explanation for why distillation can hurt performance on rare subgroups in certain settings (§3.8), building on the self-distillation analysis for kernel methods developed in Mobahi et al. (2020).

(iv) we propose two simple modifications of distillation that mitigate the above problem, based on applying per-subgroup mixing weights and margins (§4); these perform well empirically (§5).

At the outset, we note that while the paper's analysis is primarily (though not exclusively) empirical, our focus is on systematic *analysis* aiming at understanding a non-trivial phenomenon, rather than merely empirical *comparison*. Specifically, §3 is devoted to carefully understanding the extent of, and causes for, the non-uniform gains of distillation. This is in line with works which empirically analyse neural network phenomena (Zhang et al., 2017; Müller et al., 2019; Nakkiran et al., 2020; Neyshabur et al., 2020).

## 2 Background and related work

**Knowledge distillation**. Consider a multi-class classification problem over instances $\mathcal{X}$ and labels $\mathcal{Y} = [L] \doteq \{1, \ldots, L\}$. Given a training set $S = \{(x_n, y_n)\}_{n=1}^N$ drawn from some distribution $\mathbb{P}$, we seek a classifier $h: \mathcal{X} \to \mathcal{Y}$ that minimises the *misclassification error* $E_{\mathrm{avg}}(h) \doteq \mathbb{P}(h(x) \neq y)$. In practice, one may learn *logits* $f: \mathcal{X} \to \mathbb{R}^L$ to minimise $\hat{R}(f) = \frac{1}{N} \sum_{n=1}^N \ell(y_n, f(x_n))$, where $\ell$ is a loss function such as the softmax cross-entropy, which for softmax probabilities $p_y(x) \propto \exp(f_y(x))$ is $\ell(y, f(x)) \doteq -\log p_y(x)$. One may then classify the sample via $h(x) = \arg\max_{y \in [L]} f_y(x)$.

Knowledge distillation (Bucilă et al., 2006; Hinton et al., 2015) employs the logits $f^{\mathrm{t}}: \mathcal{X} \to \mathbb{R}^L$ of a "teacher" model to train a "student" model $f^{\mathrm{s}}: \mathcal{X} \to \mathbb{R}^L$, via minimising

$$\hat{R}_{\mathrm{dist}}(f) = \frac{1}{N} \sum_{n=1}^N \left[ (1 - \alpha) \cdot \ell(y_n, f(x_n)) + \alpha \cdot \sum_{y' \in [L]} p_{y'}^{\mathrm{t}}(x_n) \cdot \ell(y', f(x_n)) \right], \quad (1)$$

where $\alpha \in [0, 1]$. Here, one converts the teacher logits to probabilities $p^{\mathrm{t}}: \mathcal{X} \to \Delta_L$ for simplex $\Delta$, e.g. via a softmax transformation $p_{y'}^{\mathrm{t}}(x) \propto \exp(f_{y'}^{\mathrm{t}}(x))$. The second term *smooths* the student labels based on the teacher's confidence that they explain the sample. The first term includes the original label to prevent incorrect teacher predictions from overwhelming the student. One further important trick is *temperature scaling* of the teacher logits, so that $p_{y'}^{\mathrm{t}}(x) \propto \exp(T^{-1} \cdot f_{y'}^{\mathrm{t}}(x))$. Setting $T \gg 0$ makes $p^{\mathrm{t}}$ more uniform, thus preventing overconfident predictions (Guo et al., 2017).

We focus here on logit-based distillation, where the training data for both the teacher and the student is the same set of examples. This setup is widely considered when studying fundamental properties of distillation (Mobahi et al., 2020; Menon et al., 2021; Dao et al., 2021; Zhou et al., 2021), and is employed by the current state-of-the-art distillation technique of Beyer et al. (2021). There are other compelling setups to consider, including distillation based on transferring feature representations Heo et al. (2019) or self-supervised tasks Xu et al. (2020); Chuanguang Yang (2021); Yang et al. (2021) We leave these for future work, as the study of fairness aspects of distillation remain undeveloped even in the logit-based setting.

**Average versus subgroup performance**. The above exposition treats the misclassification error $E_{\mathrm{avg}}(h)$ as the fundamental performance measure of interest. However, suppose the data contains *subgroups* $\mathcal{G} = \{1, \ldots, G\}$. Defining the *per-subgroup errors* $\mathrm{err}_g(h) \doteq \mathbb{P}(h(x) \neq y \mid g)$, we have $E_{\mathrm{avg}}(h) = \sum_{g \in \mathcal{G}} \mathbb{P}(g) \cdot \mathrm{err}_g(h)$, which may mask errors on samples with $\mathbb{P}(g) \sim 0$ (Sagawa et al., 2020a;b; Sohoni et al., 2020). To this end, one may instead measure the *balanced* error (Menon et al., 2013) $E_{\mathrm{bal}}(h) \doteq \sum_{g \in \mathcal{G}} \frac{1}{|\mathcal{G}|} \cdot \mathrm{err}_g(h)$ which treats the subgroup distribution as uniform, or the *worst-subgroup* error (Sagawa et al., 2020a;b; Sohoni et al., 2020) $E_{\mathrm{max}}(h) \doteq \max_{g \in \mathcal{G}} \mathrm{err}_g(h)$, which focusses on the worst-performing subgroup. An intermediary is the average of the $k$ worst-performing subgroups (Williamson & Menon, 2019): for $i$th largest per-subgroup error $\mathrm{err}^{[i]}(h)$, $E_{\mathrm{top-k}}(h) \doteq \frac{1}{k} \sum_{i=1}^k \mathrm{err}^{[i]}(h)$.

The definition of $\mathcal{G}$ is a domain-specific consideration. One special case is where each label defines a subgroup (i.e., $\mathcal{G} = \mathcal{Y}$), and $\mathbb{P}(y)$ is skewed. In such *long-tail* settings (Buda et al., 2017), classifiers with good average performance can perform poorly on "tail" labels where $\mathbb{P}(y) \sim 0$.

**Related work**. There is limited prior study that dissects distillation's overall gains per subgroup. Zhao et al. (2020) showed that in incremental learning settings, distillation can be biased towards recently observed classes. We show that even in offline settings, distillation can harm certain classes. Recently, Zhou et al. (2021) studied the standard *aggregate* performance $E_{\mathrm{avg}}$ of distillation, which was tied to a certain subset of "regularisation samples". By contrast, our goal is to understand the *subgroup* performance of distillation.

| Setting | Dataset | Avg acc | Worst subgroup acc |
|---|---|---|---|
| Baseline (§3.1) | ImageNet | +0.39 | -0.43 |
| Stronger teacher (§3.2) | ImageNet | +0.17 | -1.19 |
| Long tail (§3.4) | CIFAR-100 LT | +2.17 | -1.46 |
| | ImageNet LT | +0.21 | -0.32 |
| Fairness (§3.6) | UCI Adult | +3.10 | -5.94 |
| Reduce #classes (§3.3) | CIFAR-10 | +0.55 | +0.90 |
| | CIFAR-10 LT | +1.92 | +4.40 |
| | CIFAR-100 | +1.93 | +3.33 |
| | ImageNet-100 | +0.09 | +0.06 |

Table 1: Summary of findings in the ablation analysis of distillation's subgroup performance (§3). In a range of different settings, distillation is seen to hurt the hardest subgroup accuracy (worst-$k$ class accuracy or worst subgroup according to an attribute), despite improving the average accuracy (upper rows). Decreasing number of labels helps improve the hardest classes (bottom row).

Study of the *fairness* of machine learning algorithms on under-represented data subgroups has received recent attention (Dwork et al., 2012; Hardt et al., 2016; Buolamwini & Gebru, 2018; Chzhen et al., 2019). This has prompted dissection of the performance of techniquess such as dimensionality reduction (Samadi et al., 2018), increasing model capacity (Sagawa et al., 2020a), and selective classification (Jones et al., 2021). We follow the general spirit of such works, studying a more delicate setting involving *two* separate models (the student and teacher), each with their own inductive biases. We present more discussion of related directions in §6. In a related effort, Hooker et al. (2019) explore how model compression may harm subgroup performance compared to the original model.

## 3 Are distillation's gains uniform?

We demonstrate that the gains of distillation are *not uniform across subgroups*: specifically, considering subgroups defined by classes, distillation can *harm* the student's performance on the "hardest" few classes (§3.1). To understand the genesis of this problem, we perform ablations (cf. Table 3.1) that establish its existence in settings where there are insufficient samples to model certain classes, either due to the number of classes being large (§3.3), or the class distribution being skewed (§3.4). We then identify that the student may amplify the teacher's errors (§3.7). Next, we corroborate these results for a more general notion of subgroup in a fairness dataset (§3.6). Finally, we show an analysis indicating that this behaviour is potentially a result of the teacher *confidently mispredicting* on some classes (§3.7). We conclude with formally proving that distillation can indeed harm rarest subgroups in the context of kernelized methods (§3.8).

### 3.1 Distillation hurts subgroup performance

To begin, we consider the effect of distillation on a standard image classification benchmark, namely, ImageNet. We employ a self-distillation (Furlanello et al., 2018) setup, with ResNet-34 teacher and student models, trained with standard hyperparameter choices (see Appendix B). Following Cho & Hariharan (2019), we use early stopping on the teacher model. We now ask: what is the impact of distillation on the student's *overall* and *per-class* performance? The first question has an expected answer: distillation improves the student's average accuracy by +0.4% (see the *Baseline* setting in Table 3.1). Judged by this conventional metric, distillation is thus a success.

A more nuanced picture emerges when we break down the source of the above improvement. We compute the per-class accuracies for the one-hot and distillation models, to understand how the overall gains of distillation

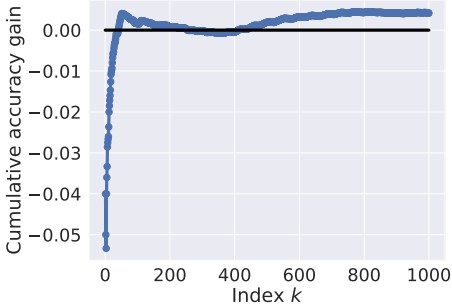

Figure 2: Cumulative gain of ResNet-34 self-distillation on ImageNet. For index $k$, we compute the gain in average accuracy over the $k$ worst classes. While average accuracy ($k = 1000$) improves by +0.4%, for $k \leq 40$, distillation *harms* over the one-hot model (evidenced by the negative gain).

| Teacher | Student | Average accuracy | Worst-10 accuracy | | Teacher | Student | Average accuracy | Worst-10 accuracy | | Teacher | Student | Average accuracy | Worst-10 accuracy |
|---------|---------|------------------|-------------------|---|---------|---------|------------------|-------------------|---|---------|---------|------------------|-------------------|
| EffnNet | Res-50 | +0.17 | -1.19 | | Res-50 | Res-50 | +0.34 | -1.58 | | Res-34 | Res-34 | +0.42 | -2.60 |
| EffnNet | Res-34 | 0.00 | -0.80 | | Res-50 | Res-34 | +0.39 | -2.05 | | Res-34 | Res-18 | +0.15 | -3.60 |
| EffnNet | Res-18 | +0.05 | -1.60 | | Res-50 | Res-18 | -0.09 | -1.00 | | Res-18 | Res-18 | -0.13 | -3.80 |

Table 2: Summary of effect of distillation on different teacher and student architectures considered for the ImageNet dataset. The comparison is with respect to the one-hot (i.e., non-distilled) student. Distillation consistently hurts accuracy of the worst 10 classes.

are distributed. Figure 2 shows that these gains are non-uniform: distillation in fact *hurts* the worst-$k$ class performance for $k \leq 40$. (See Appendix C for a detailed per-class breakdown.) Thus, distillation may harm the student on classes that it already finds difficult. At the same time we note that distillation does improve many classes, such as the classes with relatively high accuracies from the one-hot student.

Given that average accuracy improves, does it matter that performance on subgroups corresponding to the "hardest" classes suffers? While ultimately a domain-specific consideration, in general exacerbating subgroup errors may lead to issues from the fairness perspective. Indeed, we shall see that distillation can also harm in settings where the subgroups correspond to sensitive variables; see §3.6.

At this stage, it is apposite to ask whether the above is an isolated finding, or indicative of a deeper issue. We thus study each of the following in turn: (i) does the finding hold in settings beyond self-distillation? (ii) does the finding hold for other datasets, or is it simply due to the idiosyncrasies of ImageNet? (iii) what are some general characteristics of settings where the problem is manifest?

## 3.2 Is distillation biased by the model architecture?

Having begun with a self-distillation setup, we now demonstrate that similar findings hold when the student and teacher architectures differ. Continuing with the ImageNet dataset, in the second row in Table 3.1 we report statistics for the overall average accuracy and average accuracy over the worst 10 classes when distilled ResNet-50 student from a stronger teacher: Efficient-Net (Tan & Le, 2021) (more specifically, Efficient-NetV2 L) teacher achieving 85.7% accuracy. Again, we see improved average accuracy and harmed Worst subgroup accuracy, composed of the 10 classes with the lowest accuracy according to teacher's performance. In Table 2, we report statistics when varying teacher and student architectures. The harming of hard class performance under distillation holds across all scenarios: thus, our earlier results were not specific to self-distillation.

For self-distillation settings, smaller models appear to incur greater losses on the worst-class error. When distilling between different architectures (e.g., from ResNet-50 to ResNet-18), we observe that even average accuracy may not improve, as noted in Cho & Hariharan (2019). There is however no clear trend between the difference in architectures and drop in worst class performance. Finally, we note that there is little change in the teacher's and student's worst-$k$ classes; see Figure 7 (Appendix).

### 3.3 Is distillation biased by a large number of classes?

Having seen that ImageNet consistently demonstrates a performance degradation on certain classes, we now repeat the same analysis on a smaller image classification benchmark. We return to the self-distillation setup, using ResNet-56 models on CIFAR-100. On this dataset, Table 3.1 shows a (perhaps more expected) result: distillation boosts *both* the average and worst-1 class performance. This indicates that, at a minimum, the behaviour of distillation's performance gains are problem-specific; on CIFAR, distillation appears a complete win for both the average and subgroup accuracy.

One plausible hypothesis is that the tension between average and subgroup performance only manifests on problems with many labels, which might informally be considered "harder". To confirm this further while fixing the dataset, we randomly select 10% of classes from the ImageNet dataset, and only keep examples corresponding to those classes across the train and validation sets. Consistent with the results for CIFAR-100, we again find that the worst-10 class accuracy is not harmed under distillation (see the right side of Table 3.1 where we report results on ResNet-34 self-distillation).

Next, we also consider the setting where the number of train examples is fixed but the number of classes varies. To this end, we contrast results from experiments on CIFAR-10 and CIFAR-100, and CIFAR-10 LT and CIFAR-100 LT. In each of these pairs of datasets, the set of the train examples is fixed but the number of classes changes from 10 to 100. As reported in Table 3.1, we find that the accuracy over the worst 10% classes according to the teacher does not drop for neither CIFAR-10 nor CIFAR-10 LT upon distillation.

The above indicates that for problems with a few, balanced labels or with sufficiently few labels with respect there may not be a tension between average and worst-subgroup performance under distillation. However, we now show that even for problems with relatively few labels, one may harm subgroup performance if there is *label imbalance.*

### 3.4 Is distillation biased by class imbalance?

We now consider a *long-tail* setting, where the training label distribution $\mathbb{P}(y)$ is highly non-uniform. Following the long-tail learning literature (Cui et al., 2019; Cao et al., 2019; Kang et al., 2020), we construct "long-tailed" (**LT**) versions of the above datasets, wherein the training set is down-sampled so as to achieve a particular label skew. For ImageNet, we use the long-tailed version from Liu et al. (2019). For other datasets, we down-sample labels to follow $\mathbb{P}(y = i) \propto \frac{1}{\mu^i}$ for constant $\mu$ and $i \in [L]$ (Cui et al., 2019). The ratio of the most to least frequent class is set to 100.

From the *Long tail* setting in Table 3.1, we note that on both CIFAR-100-LT and ImageNet-LT, accuracy over the hardest classes drops (we report results from self-distillation using ResNet-56 for CIFAR-100 and ResNet-50 ImageNet). The former is particularly interesting, given that the standard CIFAR-100 shows gains amongst the hardest classes. This provides evidence that for some "harder" problems — e.g., where there are insufficiently many samples from which to model a particular class — there may be a tension between the average and subgroup performance.

### 3.5 Is distillation biased by hyperparameter choice?

We study the impact of hyper parameters on the bias of distillation in the case of CIFAR-100 LT and present the results in Figure 3. We find that, across student architectures, for all temperatures and choices for the mixing parameter $\alpha$, distillation improves the average accuracy but harms the worst-10 class accuracy.

### 3.6 Beyond classes: other choices of subgroups

Our analysis thus far has focused on subgroups defined by classes. This choice is natural for long-tailed problems, where it is important to ensure good model performance on rare classes (Kang et al., 2020). In other problems, different choices of subgroups may be appropriate. For example, in problems arising in fairness, one may define subgroups based on certain sensitive attributes (e.g., sex, race). In such settings, does one similarly see varying gains from distillation across subgroups?

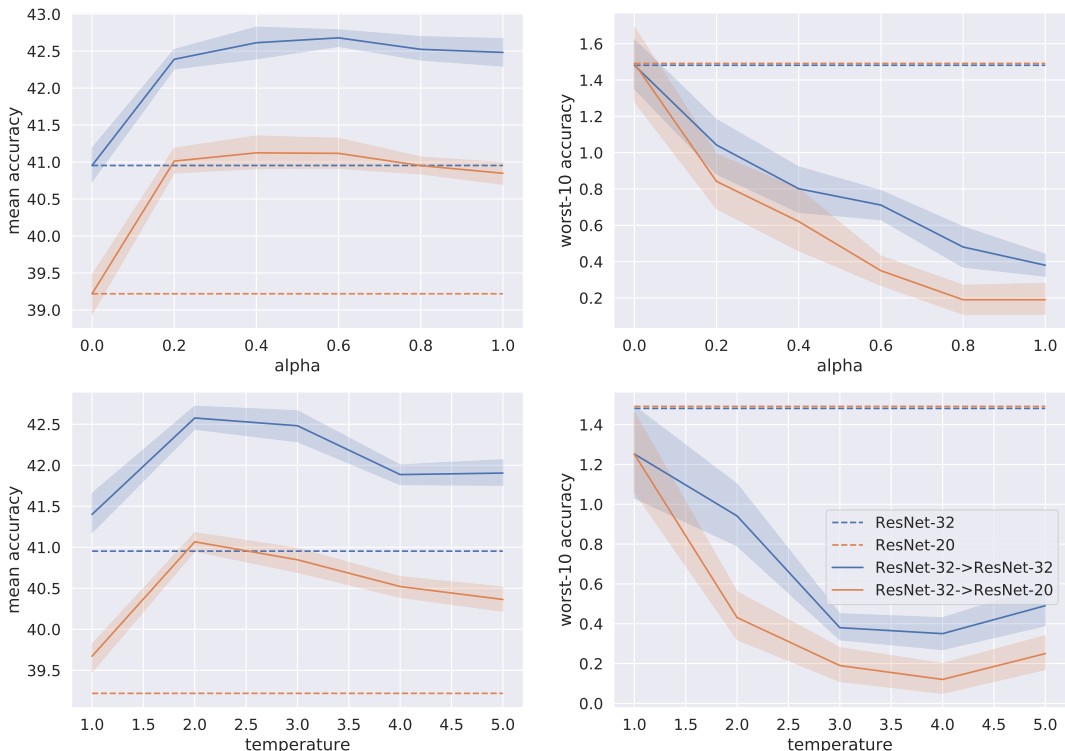

Figure 3: Impact of hyper parameters on the bias of distillation in the case of CIFAR-100 LT. We find that, across student architectures, for all temperatures and choices for the mixing parameter $\alpha$, distillation improves the average accuracy but harms the worst-10 class accuracy.

We confirm this can indeed hold on the UCI Adult dataset using random forest models (details in Appendix C.3). This data involves the task of predicting if an individual's income is $\geq 50K$ or not, and possesses subgroups defined by the individual's race and sex. Akin to the preceding results, we find that distillation can significantly improve *overall* accuracy, at the expense of *degrading* accuracy on certain rare subgroups, e.g., Black women; see Table 3.1, and Table 9 (Appendix). This further corroborates our basic observation on the non-uniform distribution of distillation's gains.

A distinct notion of subgroup was recently considered in Zhou et al. (2021), who identified the impact of certain "regularisation samples" on distillation. These are a subset of training samples which were seen to degrade the *overall* performance of distillation. It is of interest whether such a subgroup relates to our previously studied subgroups of "hard" classes; e.g., is there an abundance of regularisation samples in such subgroups, which might explain the poor performance of distillation? In Appendix C.4, we study the relationship between regularisation samples, and the per-label subgroups from our analysis; we find that, in general, these may be complementary notions.

### 3.7 Why distillation hurts subgroups: a margin view

The above has established that in a range of scenarios, distillation can hurt performance on subgroups defined by classes. However, a firm understanding of *why* this happens remains elusive. To study this, we consider ResNet-56 self-distillation on CIFAR-100-LT — which showed a stark gap between the average and subgroup (i.e., worst-1 class) performance — and dissect the logits of the teacher and distilled student. (See the Appendix for plots where the teacher and student architectures differ.) Across classes, we seek to understand: (i) how aligned are the student and teacher *accuracies*? (ii) how reliable are the models' *probability estimates*? (iii) how do the models' *confidences* behave?

For a *test*[1] example $(x, y)$ and predicted label distribution $p(x) \in \Delta_L$, we thus compute each models' accuracy, log-loss $\ell_{\log}(y, p(x)) = -\log p_y(x)$, and *margin* (Koltchinskii & Panchenko, 2002) $\ell_{\text{marg}}(y, p(x)) = p_y(x) - \max_{y' \neq y} p_{y'}(x)$. Note that the latter may be negative if the model predicts the incorrect label for the example. Figure 4 shows these metrics on 10 class buckets: these are created by sorting the 100 classes according to the teacher accuracy, and then creating 10 buckets of classes. Within each bucket, we compute the average of the metric.

Remarkably, for 5 out of 10 class buckets, average margins are *negative*, suggesting that the teacher is often *wrong yet confident* in predicting these classes. Here, the student accuracy generally worsens compared to the teacher. Further, log-loss increases across all buckets (including those where accuracy improves), indicating reduced confidence in the true class of the distilled student. This points at a potential source of the poorer performance on the worst-1 accuracy. Recall that the distilled student's aim is to mimic the teacher's logits on the *training* samples. This is a proxy to the student's true goal, which is mimicking these logits on *test* samples, so as to generalise similar to the teacher. When such generalisation happens, the student can be expected to roughly inherit the teacher's per-class performance; in settings like the above, this unfortunately implies poor performance on classes with negative teacher margin.

### 3.8 Why distillation can hurt subgroups: theory

To complement the preceding empirical analysis, we now seek to theoretically establish certain conditions under which distillation can *provably* harm performance on subgroups. We build upon the recent analysis of self-distillation for kernelised models in Mobahi et al. (2020). Specifically, consider the ridge regression problem:

$$\min_{f_{\mathbf{w}} \in \mathcal{F}} \sum_{n=1}^{N} (f_{\mathbf{w}}(\mathbf{x}_n) - z_n)^2 + \lambda \|\mathbf{w}\|_2^2, \tag{2}$$

where $\mathcal{F} = \{f_{\mathbf{w}} \colon \mathbf{x} \mapsto \mathbf{w}^\top \mathbf{x} \mid \mathbf{w} \in \mathbb{R}^D\}$, and $\{z_n\}_{n=1}^N$ is a set of target labels. Let $f^*$ denote the minimizer of (2) where each $z_n = y_n$. Similarly, let $f^{\text{dist}}$ denote the minimizer where each $z_n = f^*(x_n)$, i.e., the self-distillation solution. Mobahi et al. (2020) showed how such self-distillation can be interpreted as a regulariser, in the sense of $f^{\text{dist}}$ implicitly down-weighting low variance directions in the data.

Our analysis holds for the following data distribution: suppose we have data from $L$ subgroups, and targets $y_n \in \mathbb{R}$. Let $\{\mathbf{o}_i\}_{i \in [L]}$ be $L$ orthonormal vectors in $\mathbb{R}^D$, corresponding to the input representation $\mathbf{x}_i$ from different subgroups. Let $\mathcal{M}_i$ denote all samples with subgroup $i$, and $d_i = |\mathcal{M}_i|$. Let $\epsilon_i = \frac{1}{d_i \cdot \bar{y}_i^2} \sum_{k \in \mathcal{M}_i} (f^*(x_k) - y_k)^2$ denote the mean error of examples from subgroup $i$, and $\epsilon_i^{\text{dist}} = \frac{1}{d_i \cdot \bar{y}_i^2} \sum_{k \in \mathcal{M}_i} (f^{\text{dist}}(x_k) - y_k)^2$ the mean error from subgroup $i$ of the distilled model. Here, the error is normalised by the mean subgroup target $\bar{y}_i^2 = \frac{1}{d_i} \sum_{k \in \mathcal{M}_i} y_k^2$.

**Theorem 1.** *Let the subgroups be in decreasing order according to their frequency with $i < j$ implying $d_i > d_j$. For any $1 \leq i < j \leq L$, if $\lambda < \sqrt{d_i d_j}$, then,*

$$\epsilon_j - \epsilon_j^{\text{dist}} \leq \epsilon_i - \epsilon_i^{\text{dist}}.$$

Theorem 1 shows that distillation has a worse effect on subgroup $j$ compared to $i$. Thus, the less represented directions of the data (e.g. those corresponding to the rare subgroups) would suffer more from the induced regularisation, as long as the regularisation strength $\lambda$ is sufficiently small. This is consistent with the observation of hardest classes (often the rarest classes) suffering the most from distillation (and in particular, self-distillation, as we extensively study in §5).

### 3.9 Discussion

We have found that distillation's *gains* in average accuracy can be at the expense of *degradation* in "hard" class accuracy, which can *amplify* the gap between worst- and average-class performance. Given a widespread

---

[1]The choice of test, rather than train, example is crucial: an overparameterised teacher will likely correctly predict *all* training samples, thus rendering the above statistics of limited use. To leverage the insights from the above analysis in practice, we shall use a holdout set that can be carved out from the training set.

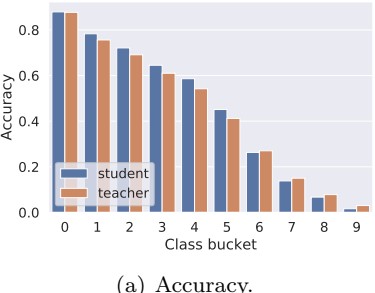

(a) Accuracy.

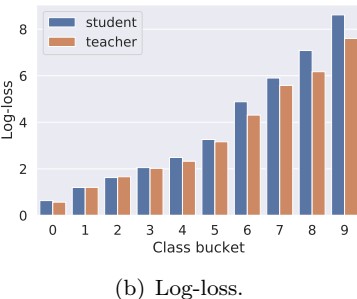

(b) Log-loss.

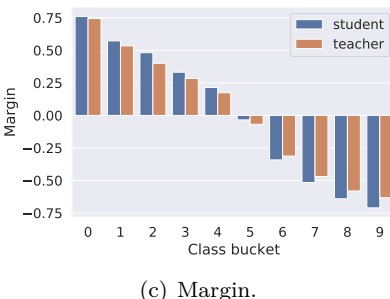

(c) Margin.

Figure 4: Logit statistics on CIFAR-100 LT, for the teacher and distilled student under a self-distillation setup (ResNet-56 → ResNet-56). We show the statistics on 10 class buckets: these are created by sorting the 100 classes according to the teacher accuracy, and then creating 10 groups of classes. As expected, the student follows the general trend of the teacher model. Strikingly, we observe that the teacher model tends to systematically *confidently mispredict* samples in the higher buckets, thus incurring a *negative* margin; such misplaced confidence is largely transferred to the student, whose accuracy suffers on such buckets. Note that we consider statistics on the *test* set.

use of distillation, we believe this systematic drop in worst class accuracy is of interest. Further, we emphasise that this is a more worrisome phenomenon than distillation potentially not helping worst class as much as it helps the easiest classes.

## 4 Improving subgroup performance via adaptive distillation methods

We now study simple means of correcting distillation to prevent the degradation of subgroup performance. These leverage the insight that the behaviour is potentially a result of the teacher *confidently mispredicting* on some subgroups. In the following, for concreteness and simplicity, we focus on subgroups that are given by the individual classes.

### 4.1 Distillation with adaptive mixing weights

In §3.7, we saw that distillation can hurt on classes where the teacher is inherently inaccurate. Such inaccuracy may in fact be *amplified* by the student, which is hardly desirable. An intuitive fix is to simply rely less on the teacher for classes where it performs poorly, or is otherwise not confident; instead, the student can simply fall back onto the one-hot training labels themselves. Formally, for *per-class* mixing weights $(\alpha_1, \ldots, \alpha_L) \in [0, 1]^L$, the student can minimise

$$\bar{R}_{\text{dist}}(f) = \frac{1}{N} \sum_{n=1}^{N} \left[ (1 - \alpha_{y_n}) \cdot \ell(y_n, f(x_n)) + \alpha_{y_n} \cdot \sum_{y' \in [L]} p_{y'}^{\text{t}}(x) \cdot \ell(y', f(x_n)) \right]. \tag{3}$$

This objective introduces a mixing weight $\alpha_y$ per-class, which allows us to weigh between teacher predictions and one-hot labels for each class independently. By contrast, in the standard distillation setup equation 1 we only have a single weight $\alpha$ that is common for all classes.

How do we choose the weights $\alpha_y$? In the standard distillation objective equation 1, one only needs to tune a single scalar $\alpha$, which is amenable to, e.g., cross-validation. By contrast, equation 3 involves a single scalar for each label, which makes any attempt at grid search infeasible. Following the observations in §3.7, we propose the following intuitive setting of $\alpha_y$ given teacher predictions $p^{\text{t}}$:

$$\alpha_y = \max\left(0, \mathbb{E}_{x|y}\left[\gamma_{\text{avg}}(y, p^{\text{t}}(x))\right]\right) \tag{4}$$

$$\gamma_{\text{avg}}(y, p^{\text{t}}(x)) \doteq p_y^{\text{t}}(x) - \frac{1}{L-1} \sum_{y' \neq y} p_{y'}^{\text{t}}(x). \tag{5}$$

Equation 4 places greater faith in the teacher model for those classes which it predicts *correctly* with *confidence*, i.e., with large average margin $\gamma_{\mathrm{avg}}$. When this margin is *negative* — so that the teacher is *incorrect* on average, which can occur on classes that are rare in the training set — we set $\alpha_y = 0$, and completely ignore the teacher predictions.

The above requires estimating the expectation $\mathbb{E}_{x|y}[\cdot]$, which requires access to a labelled sample. This may be done using a holdout set, which we follow in our experiments. It is worth noting the limitations of such an approach. First, such a scheme is, of course, not feasible in settings where the teacher model is used to label a large unlabelled pool of samples. While this is an important practical setting, further study of alternate means of mitigating the amplification of biases is required. Further, when certain data subgroups only have very few associated samples, it may not be feasible to hold out a part of the data. This issue also affects popular approaches for mitigating subgroup bias, e.g., Sagawa et al. (2020a). It would be of interest for future work to develop methods not requiring a holdout set for hyperparameter estimation. Nonetheless, in the next section we report improvements from such a scheme even for long tail datasets.

## 4.2 Distillation with per-class margins

Our second approach for improving distillation on harder classes is to leverage recent developments in *long-tail learning*, where the goal is to improve performance on rare classes. Specifically, Menon et al. (2020) proposed a variant of the softmax cross-entropy with *label margins* $\rho_{yy'}$:

$$\ell(y, f(x)) = \log\left[1 + \sum_{y' \neq y} \rho_{yy'} \cdot e^{f_{y'}(x) - f_y(x)}\right]. \tag{6}$$

Intuitively, this penalises predicting label $y'$ instead of $y$ when $\rho_{yy'}$ is large. For training label distribution $\pi$, Cao et al. (2019) proposed to set $\rho_{yy'} \propto \exp(\pi_y^{-1/4})$, so that rare labels receive a higher weight when misclassified. Khan et al. (2018); Ren et al. (2020); Menon et al. (2020); Wang et al. (2021) showed gains with $\rho_{yy'} \propto \frac{\pi_{y'}}{\pi_y}$, so that rare labels are not confused with common ones.

We adapt such techniques to our setting, with the intuition that we ought to increase the student penalty for misclassifying those "hard" classes that the teacher has difficulty modeling. We thus choose $\rho_{yy'} = \frac{\alpha_{y'}}{\alpha_y}$, where $\alpha_y$ is the adaptive per-class mixing weight from the previous section. This discourages the model from confusing "hard" labels $y$ with "easy" labels $y'$, when $\alpha_{y'} > \alpha_y$. To avoid a division by 0 issue, we add a small offset to $\alpha_y$ when it becomes 0.

We may understand the effect of Equation 6 by studying how it impacts the *Bayes-optimal* student model predictions, i.e., the optimal predictions in the infinite sample limit, and without a model capacity restriction. We have the following.

**Lemma 2.** *Let $\ell$ be per (6), with $\rho_{yy'} \doteq \frac{\alpha_{y'}}{\alpha_y}$. Let $f^*$ be the minimiser of $R_{\mathrm{dist}}(f) \doteq \mathbb{E}_x(p^{\mathrm{t}}(x))^\top \ell(f(x))$. Then, $\forall x \in \mathcal{X}, y \in [L]$, $f_y^*(x) = \log \frac{p_y^{\mathrm{t}}(x)}{\alpha_y}$.*

Lemma 2 illustrates that using per-class margins encourages the student to mimic the teacher predictions $p^{\mathrm{t}}(x)$, but with an important modification: we up-weight the probabilities for classes that the teacher does poorly on ($\alpha_y \sim 0$). Intuitively, this makes it easier for the student to improve performance on classes with small teacher margin.

## 4.3 Relation to existing work

Previous works varied distillation supervision across examples towards improving *average* accuracy. Proposals included weighting samples based on the ratio (Tang et al., 2019; Zhou et al., 2021), and difference (Zhang et al., 2020) between student and teacher score. Similarly, Zhou et al. (2020b) proposed to only apply distillation on samples the teacher gets correct. In our experiments, we compare against the baseline from Zhou et al. (2021), weighting examples based on teacher and student scores.

Table 3: Summary of student's average accuracy using one-hot and distilled labels. *Worst k* denotes accuracy over the worst $k$ classes. Global and adaptive temperatures $\alpha_y$ selected using a held out dev set. The proposed AdaAlpha technique improves worst class accuracy over vanilla distillation. For AdaMargin on CIFAR-100 LT we observed divergence during training, presumably due to this method being sensitive to the selection of hyperparameters, which in turn are estimated on very small number of examples per class.

| Dataset | Method | Per-class accuracy statistics | | |
| --- | --- | --- | --- | --- |
| | | Mean | Worst-1 | Worst-10 |
| CIFAR-100 | One-hot | $73.31 \pm 0.10$ | $45.67 \pm 0.94$ | $52.12 \pm 0.47$ |
| | Distillation | $75.24 \pm 0.10$ | $49.00 \pm 1.63$ | $54.65 \pm 0.79$ |
| | AdaAlpha | $75.43 \pm 0.31$ | $49.33 \pm 0.94$ | $56.42 \pm 0.88$ |
| | AdaMargin | $75.15 \pm 0.30$ | $\mathbf{51.33} \pm 0.47$ | $\mathbf{56.62} \pm 0.35$ |
| ImageNet | One-hot | $75.94 \pm 0.05$ | $12.00 \pm 1.63$ | $21.88 \pm 0.76$ |
| | Distillation | $76.14 \pm 0.10$ | $14.00 \pm 1.41$ | $21.54 \pm 0.58$ |
| | AdaAlpha | $\mathbf{76.15} \pm 0.16$ | $\mathbf{15.50} \pm 2.18$ | $\mathbf{23.07} \pm 0.95$ |
| | AdaMargin | $76.08 \pm 0.17$ | $15.00 \pm 1.00$ | $22.42 \pm 1.06$ |

| Dataset | Method | Per-class accuracy statistics | | | |
| --- | --- | --- | --- | --- | --- |
| | | Mean | Worst-1 | Worst-10 | Worst-100 |
| CIFAR-100 LT | One-hot | $43.22 \pm 0.05$ | $0.00 \pm 0.00$ | $2.33 \pm 0.40$ | N/A |
| | Distillation | $45.17 \pm 0.15$ | $0.00 \pm 0.00$ | $1.80 \pm 0.03$ | N/A |
| | AdaAlpha | $\mathbf{48.57} \pm 0.11$ | $\mathbf{0.67} \pm 0.47$ | $\mathbf{4.20} \pm 0.24$ | N/A |
| | AdaMargin* | | Training diverges | | |
| ImageNet LT | One-hot | $45.71 \pm 0.14$ | $0.00 \pm 0.00$ | $0.00 \pm 0.00$ | $0.68 \pm 0.12$ |
| | Distillation | $45.78 \pm 0.08$ | $0.00 \pm 0.00$ | $0.00 \pm 0.00$ | $0.58 \pm 0.02$ |
| | AdaAlpha | $45.90 \pm 0.18$ | $0.00 \pm 0.00$ | $0.00 \pm 0.00$ | $0.83 \pm 0.09$ |
| | AdaMargin | $45.98 \pm 0.25$ | $0.00 \pm 0.00$ | $0.00 \pm 0.00$ | $\mathbf{0.85} \pm 0.11$ |

Other techniques modifying distillation towards improved *average* accuracy include: going beyond logit matching and distilling intermediate layer representations (Li et al., 2019; Sun et al., 2020), introducing auxiliary teachers of intermediate capacity for bridging the gap between the teacher and the student (Mirzadeh et al., 2020), distilling using a large unlabel dataset annotated by the teacher (Cotter et al., 2021), or training the student and the teacher jointly (Zhou et al., 2017). It will be of interest for future work to understand the impact of these techniques on the *worst-class* performance of the student model.

In the special case where subgroups are defined by classes, and classes have an inherent skew, methods from the class-imbalance or long-tail learning literature are also relevant. Prominent strategies to address this problem include techniques that modify the *sampling distribution* (Kubat & Matwin, 1997; Wallace et al., 2011; Mikolov et al., 2013; Mahajan et al., 2018; Yin et al., 2018; Iscen et al., 2021), adjust the *decision threshold* post-hoc (Fawcett & Provost, 1996; Provost, 2000; Maloof, 2003; King & Zeng, 2001; Collell et al., 2016), adjust the *classifier weights* post-hoc (Kim & Kim, 2019; Kang et al., 2020), modify the *loss function* (Xie & Manski, 1989; Morik et al., 1999; Cui et al., 2019; Zhang et al., 2017; Cao et al., 2019; Tan et al., 2020; Ren et al., 2020; Menon et al., 2020; Wang et al., 2021), perform *data-augmentation* (Chawla et al., 2002; Müller et al., 2019; Chu et al., 2020; Temraz & Keane, 2022; Zada et al., 2022), *ensemble* the base classifier (Fan et al., 1999; Chawla et al., 2003; Galar et al., 2012; Sharma et al., 2020; Zhou et al., 2020a), perform *reinforcement-learning* (Fan et al., 2021), and attach exits for weighting examples by how confidently they are predicted by the intermediate layers (Duggal et al., 2020). We refer the reader to Zhang et al. (2021) for a more comprehensive survey of recent works.

While the above techniques are typically devised in a non-distillation setup, one may nonetheless consider adapting them when training the student model; e.g., one may ensemble many distilled student models in hopes of reducing any degradation on tail classes. We emphasise however that the long-tail learning setup is only *one instance* of the problem considered in this work, wherein the data comprises subgroups that *may have perfectly balanced frequencies* (e.g., the standard CIFAR and ImageNet datasets considered above). Many of the above strategies are not immediately applicable in such cases (e.g., strategies that require adjusting the sampling distribution based on frequency would correspond to standard uniform sampling).

## 5 Results for adaptive distillation methods

We now present results that further corroborate the potential non-uniform gains of distillation, and the ability to mitigate this with the techniques of the previous section. We emphasise here that our goal is expressly *not* to improve over the state-of-the-art in distillation techniques; rather, we wish to verify the key principles identified in the preceding study, which considers distillation from a novel angle (i.e., in terms of subgroup rather than average performance).

Table 4: Summary of student's accuracy using one-hot and distilled labels across teacher and student architectures on the CIFAR-100 LT dataset. *W-k* denotes accuracy over the worst *k* classes. We find that the proposed methods improve worst class performance in all cases.

| Teacher | Student | Method | Per-class accuracy statistics | | |
|---|---|---|---|---|---|
| | | | Mean | W-10 | W-20 |
| ResNet-56 | ResNet-14 | One-hot | 39.95 | 0.80 | 3.30 |
| | | Distillation | 39.53 | 1.30 | 3.35 |
| | | AdaAlpha | 40.34 | 2.00 | 4.00 |
| | | AdaMargin | **40.80** | **2.40** | **4.80** |
| ResNet-32 | ResNet-14 | One-hot | 39.95 | 0.80 | 3.30 |
| | | Distillation | 39.84 | 1.20 | 3.00 |
| | | AdaAlpha | **40.21** | 1.70 | 3.95 |
| | | AdaMargin | 40.16 | **2.20** | **4.80** |
| ResNet-14 | ResNet-14 | One-hot | 39.95 | 0.80 | 3.30 |
| | | Distillation | 40.51 | 0.00 | 2.05 |
| | | AdaAlpha | 40.79 | 0.80 | 2.45 |
| | | AdaMargin | **41.26** | **1.20** | **3.70** |

Table 5: Comparing the Ada* approaches against Zhou et al. (2021): a method weighting examples in distillation based on teacher and student scores. *Worst-k* denotes accuracy over the worst *k* classes. We find that Zhou et al. (2021) is competitive in terms of improving the mean accuracy, but, contrary to the Ada* methods, does not mitigate the issue of distillation harming accuracy over the worst classes.

| Dataset | Method | Per-class accuracy statistics | | | |
|---|---|---|---|---|---|
| | | Mean | Worst-1 | Worst-10 | Worst-100 |
| CIFAR-100 | One-hot | $73.31 \pm 0.10$ | $45.67 \pm 0.94$ | $52.12 \pm 0.47$ | N/A |
| | Distillation | $75.24 \pm 0.10$ | $49.00 \pm 1.63$ | $54.65 \pm 0.79$ | N/A |
| | Zhou et al. (2021) | $\mathbf{75.74} \pm 0.20$ | $50.33 \pm 0.47$ | $54.93 \pm 0.34$ | N/A |
| | AdaAlpha | $75.43 \pm 0.31$ | $49.33 \pm 0.94$ | $56.42 \pm 0.88$ | N/A |
| | AdaMargin | $75.15 \pm 0.30$ | $\mathbf{51.33} \pm 0.47$ | $\mathbf{56.62} \pm 0.35$ | N/A |
| ImageNet LT | One-hot | $45.71 \pm 0.14$ | $0.00 \pm 0.00$ | $0.00 \pm 0.00$ | $0.68 \pm 0.12$ |
| | Distillation | $45.78 \pm 0.08$ | $0.00 \pm 0.00$ | $0.00 \pm 0.00$ | $0.58 \pm 0.02$ |
| | Zhou et al. (2021) | $\mathbf{46.02} \pm 0.13$ | $0.00 \pm 0.00$ | $0.00 \pm 0.00$ | $0.56 \pm 0.00$ |
| | AdaAlpha | $45.90 \pm 0.18$ | $0.00 \pm 0.00$ | $0.00 \pm 0.00$ | $0.83 \pm 0.09$ |
| | AdaMargin | $45.98 \pm 0.25$ | $0.00 \pm 0.00$ | $0.00 \pm 0.00$ | $\mathbf{0.85} \pm 0.11$ |

**Setup**. We report results on the datasets used in §3: CIFAR-100, ImageNet; and long-tailed (**LT**) versions of the same. For brevity, we report results under self-distillation. (For results with varying architectures, see the Appendix.) Thus, for each dataset, we train a one-hot teacher ResNet model, which is distilled to a student ResNet of the same depth. We use ResNet-56 models for CIFAR, and ResNet-50 models for all other datasets. We employ the same hyper-parameters as used in §3, except we use non-early stopped teachers for consistency across datasets; see the Appendix for details.

We compare: (i) one-hot training of the student (ii) standard distillation, i.e., minimising Equation 1 (iii) **AdaAlpha**, our proposed distillation objective with adaptive mixing between one-hot and teacher labels Equation 3, and $\alpha$ as per Equation 4 (iv) **AdaMargin**, our proposed distillation objective with adaptive margins (Equation 6), and $\rho_{yy'} = \frac{\alpha_{y'}}{\alpha_y}$. For each method, we report: (i) the standard *mean* accuracy

over all classes; (ii) the accuracy over the *worst*-1 class; and (iii) the mean accuracy over *worst*-10 (and *worst*-100 for the LT datasets) classes.

For the Ada-* methods, per §4, creating the label-dependent $\alpha_y$ requires estimating the teacher's generalisation performance. To do this, we create a random holdout split of the training set. For non-LT datasets, we randomly split into 80% (new train) – 20% (dev). For LT datasets, for each class we hold out $k$ examples into the dev set ($k = 50$ for Imagenet-LT, $k = 20$ for CIFAR-100-LT), or half of examples for a class if the total number of per class examples is $\leq 2k$. We train an initial teacher on the new train slice of data, and estimate its per-class performance on the holdout dev slice. We then estimate $\alpha_y$ as per, e.g., Equation 4.

Table 3 summarises the results for all methods. We make the following observations.

**AdaAlpha improves mean accuracy over distillation**. The proposed AdaAlpha method consistently improves standard mean accuracy over vanilla distillation. Thus, AdaAlpha does not sacrifice distillation's gains on average class performance, which is desirable. We note that while the improvement is not significant for most scenarios, our main goal is in improving the worst-accuracy; thus, even a slight improvement in mean accuracy is an additional desirable outcome. Other techniques sometimes perform slightly worse than standard distillation on this metric; however, as we now see, this is compensated by gains on other important dimensions.

**AdaAlpha improves worst-accuracy over distillation**. The proposed method consistently improves the worst-class accuracy compared to standard distillation: thus, the technique largely fulfills their design goal of improving performance on "hard" classes. Contrasting this observation, notice on ImageNet-LT dataset how Zhou et al. (2021) can improve *average* accuracy, obtaining the best mean accuracy, however at the cost of worsened *worst-classes* accuracy, as shown in Table 5. Finally, we notice that the improvement on the worst classes is much more significant than on average for AdaAlpha. This is expected and desirable, as we set out to target the former in this work. Significantly improving both the worst and average class accuracies would be a desirable future work.

**Comparison of AdaAlpha and AdaMargin**. In the Appendix, we report per class statistics for CIFAR-100 LT. AdaMargin flattens both the margin and log-loss distributions, reducing confidence on the poorly classified, tail classes. AdaAlpha consistently increases log-loss across classes, and improves margins on few buckets, leading to a positive margin on one bucket where all other methods give negative margins. Intuitively, AdaMargin tries to more aggressively control the worst-class accuracy; when this succeeds, there is a large payoff, but there is also greater risk of overfitting. In Appendix C.6, we confirm that the success of AdaAlpha is not immediately replicated by simpler baselines such as shuffling temperatures and removing distillation from the hardest labels.

**Varying teacher and student architectures** In Table 4 we vary the teacher and student architectures in the distillation experiments on the CIFAR-100 LT dataset. We find that the Ada* methods improve the worst class performance in all cases. Particularly the AdaMargin method proves effective, and improves both the average and the worst-10 accuracy by the highest margin.

# 6 Discussion and future work

Our goal of ensuring equitable performance across classes can be seen as encouraging *fairness* across subgroups defined by the classes. This is subtly different to the classical fairness literature (Calders & Verwer, 2010; Dwork et al., 2012; Hardt et al., 2016), wherein the subgroups are defined by certain *sensitive attributes*. Broadly, fairness techniques attempt to learn models that predict the target label *accurately*, but the subgroup label *poorly*; these are inadmissible for our setting, wherein the two labels exactly coincide. Ensuring fairness across subgroups defined by the classes has been studied in Mohri et al. (2019); Williamson & Menon (2019); Sagawa et al. (2020a). Adapting such algorithms to the distillation setting is of interest for future work. More broadly, furthering the understanding of when distillation can hurt under-represented subgroups is of interest.

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

# A Proofs

*Proof of Lemma 2.* We may write

$$\ell(y, f(x)) = \log\left[1 + \sum_{y' \neq y} \frac{\alpha_{y'}}{\alpha_y} \cdot e^{f_{y'}(x) - f_y(x)}\right]$$

$$= \log\left[1 + \sum_{y' \neq y} e^{\ln\alpha_{y'} - \ln\alpha_y} \cdot e^{f_{y'}(x) - f_y(x)}\right]$$

$$= \log\left[1 + \sum_{y' \neq y} e^{\bar{f}_{y'}(x) - \bar{f}_y(x)}\right]$$

$$= -\log\frac{e^{\bar{f}_y(x)}}{\sum_{y' \in [L]} e^{\bar{f}_{y'}(x)}},$$

where $\bar{f}_y(x) \doteq f_y(x) + \ln\alpha_y$. The population loss is

$$R_{\text{dist}}(f) = \mathbb{E}_x\left[(p^{\text{t}}(x))^\top \ell(f(x))\right]$$
$$= \mathbb{E}_x\left[\text{KL}(p^{\text{t}}(x) \,\|\, \bar{p}^{\text{s}}(x))\right],$$

where $\bar{p}_y^{\text{s}}(x) \propto \exp(\bar{f}_y(x))$. Thus, at optimality we must have $\bar{p}^{\text{s}}(x) = p^{\text{t}}(x)$, or $\bar{f}_y(x) = \log p_y^{\text{t}}(x)$. By definition of $\bar{f}$, we thus see that $f_y(x) = \log p_y^{\text{t}}(x) - \log\alpha_y = \log\frac{p_y^{\text{t}}(x)}{\alpha_y}$. $\qquad\square$

# B Details of experiments

## B.1 Architecture

We use ResNet with batch norm (He et al., 2016) for all our experiments with the following configurations. For CIFAR, we experiment with ResNet-56 and ResNet-32. For ImageNet, we use ResNet-50. We list the architecture configurations in terms of ($n_{\text{layer}}$, $n_{\text{filter}}$, stride) corresponding to each ResNet block in Table 6.

| Architecture | Configuration: [($n_{\text{layer}}$, $n_{\text{filter}}$, stride)] |
|---|---|
| CIFAR ResNet-32 | [(5, 16, 1), (5, 32, 2), (5, 64, 2)] |
| CIFAR ResNet-56 | [(9, 16, 1), (9, 32, 2), (9, 64, 2)] |
| ImageNet ResNet-18 | [(2, 64, 1), (2, 128, 2), (2, 256, 2), (2, 512, 2)] |
| ImageNet ResNet-34 | [(3, 64, 1), (4, 128, 2), (6, 256, 2), (3, 512, 2)] |
| ImageNet ResNet-50 | [(3, 64, 1), (4, 128, 2), (6, 256, 2), (3, 512, 2)]* |

Table 6: ResNet Architecture configurations used in our experiments (He et al., 2016). [*] Note that ImageNet ResNet-50 uses larger blocks with 3 convolutional layers per residual block compared to ResNet-18 and 34. We refer to He et al. (2016) for more details.

## B.2 Training set

For all datasets, we train using SGD and weight decay $10^{-4}$ for CIFAR, and $0.5 \times 10^{-4}$ for Imagenet datasets. We have the following dataset specific settings.

**CIFAR-100**. We train for 450 epochs with an initial learning rate of 1.0, with a linear warmup in the first 15 epochs, and an annealed learning rate schedule. We drop the learning rate by a factor of 10 at epochs number: 200, 300 and 400. We use a mini-batch size of 1024. We use SGD with Nesterov momentum of 0.9.

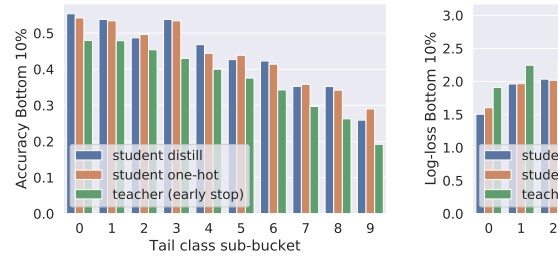 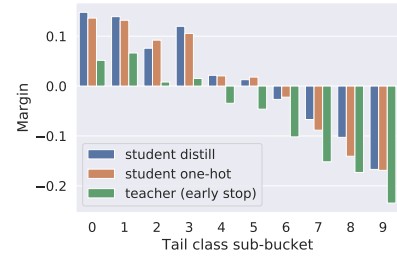

| (a) Accuracy (tail classes). | (b) Log-loss (tail classes). | (c) Margin (tail classes). |
| :---: | :---: | :---: |

Figure 5: Logit statistics for ResNet-50 self-distillation on ImageNet, for the (early-stopped) teacher, self-distilled student, and one-hot (non-distilled) student. Per Figure 4, we first create 10 class buckets. We zoom in on the "tail" bucket (comprising the 100 "hardest" classes), and further split them into 10 "tail sub-buckets". As in Figure 4, the teacher is seen to confidently mispredict most samples on the last few buckets, with such misplaced confidence being transferred to the student.

For our distillation experiments we train only with the cross-entropy objective against the teacher's logits. For each method we find the best temperature from the list of values: $\{1, 2, 3, 4, 5\}$.

**ImageNet**. We train for 90 epochs with an initial learning rate of 0.8, with a linear warmup in the first 5 epochs, and an annealed learning rate schedule. We drop the learning rate by a factor of 10 at epochs number: 30, 60 and 80. We use a mini-batch size of 1024.

For our distillation experiments we train with the distillation objective as defined in Equation 1 setting $\alpha = 0.2$. For each method we fix the temperature to 0.9.

**Long-tail (LT) datasets**. We follow setup as in the non-long tail version, except for the learning rate schedule, which we change to follow the cosine schedule (Loshchilov & Hutter, 2017).

## C    Additional Experiments

We present additional experiments to those in the body.

### C.1    Further varying datasets and model architectures

On Imagenet, we summarise statistics for *three* models: the early-stopped teacher, distilled student, and the one-hot (non-distilled) student. As with CIFAR-100-LT, we sort classes by teacher accuracy, and bucket them into 10 groups. Owing to the larger number of labels, we further zoom into the "tail" bucket (comprising the 100 "hardest" classes), and split them into 10 sub-buckets. From Figure 5, the distilled student performs worse than its one-hot counterpart on the last bucket; this is in keeping with our results in Table 3.1.

Figure 6 shows logit statistics for additional settings to considered in the body. On ImageNet-LT, e.g., we see again that the margin of the teacher model systematically worsens and becomes negative on the hardest classes.

In Table 7 we report results from the inherently long-tailed iNaturalist 2018 dataset (Van Horn & Perona, 2017). Our observations made for other considered datasets hold: adaptive margin method improves over both one hot and plain distillation in terms of the worst class accuracy. We also observe, how the average accuracy improves.

### C.2    Logit plots under Ada-* methods

Figure 8 shows the logit statistics under the proposed AdaMargin and AdaAlpha methods on CIFAR-100 LT. We see that AdaMargin can generally improve the student margin and accuracy on the hardest classes, while

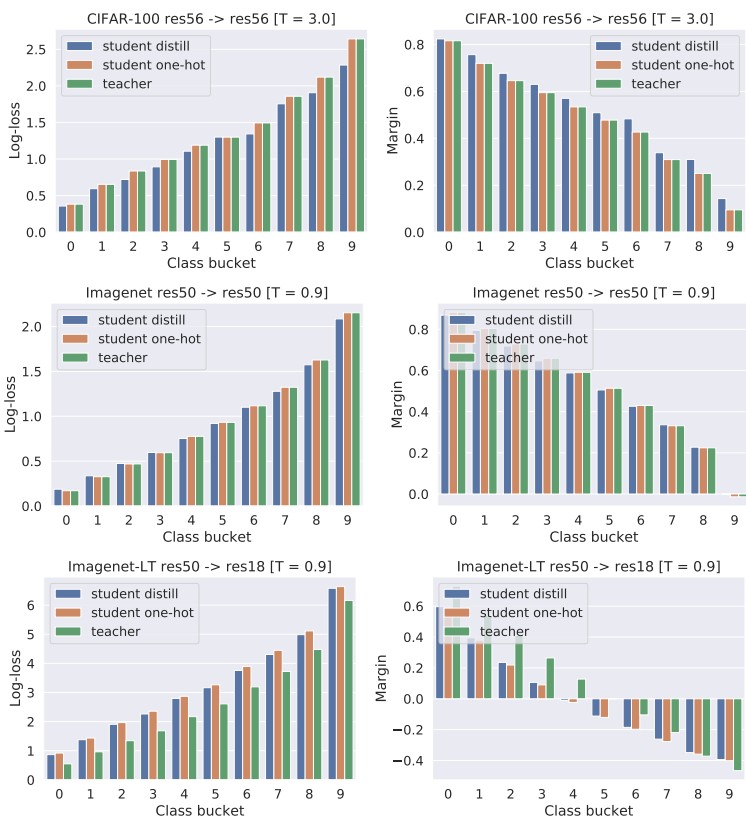

Figure 6: Logit statistics for the teacher, student with one-hot labels, and student with distilled labels across: datasets and architectures.

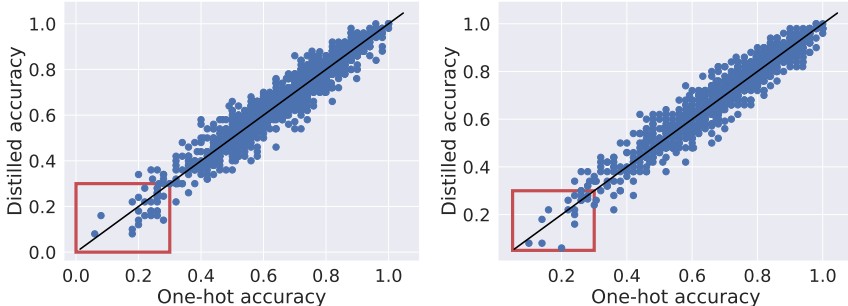

Figure 7: Per-class accuracies for one-hot and self-distilled ResNet-18 (left) and ResNet-34 (right) on ImageNet. The diagonal denotes classes where both models achieve the same accuracy. Distillation tends to worsen performance on "hard" classes for the one-hot model, i.e., those with low accuracy (red rectangle).

also reducing the log-loss. This confirms that the gains of the method come from improving behaviour of the scores on these hard classes.

## C.3 Results on Adult dataset

We report the results of an experiment on the UCI Adult dataset. This data comprises $\sim 48K$ examples, with the target being a binary label denoting whether or not an individual has income $\geq 50K$. The data is mildly imbalanced, with 24% of samples being positive.

| Method | Per-class accuracy statistics | | | |
|---|---|---|---|---|
| | **Mean** | **Worst 20** | **Top 20%** | **Δ20** |
| One-hot | 53.00 | 5.00 | 100.00 | 48.00 |
| Distill | 52.67 | 5.00 | 100.00 | 47.67 |
| AdaMargin | 53.33 | 8.33 | 98.33 | 45.00 |
| AdaAlpha avg | 54.67 | 13.33 | 100.00 | 41.33 |

Table 7: Self-distillation experiments (from Resnet-50 to Resnet-50) on the iNaturalist dataset Van Horn et al. (2018) with student's average accuracy using one-hot and distilled labels. Worst 20 denotes accuracy averaged over worst 20 classes Δ20 denotes the difference between the mean accuracy and the worst 20 classified classes. The proposed AdaMargin technique improves mean and worst-class accuracy over both one-hot training and standard distillation.

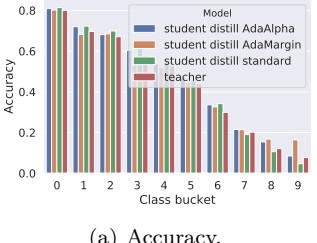
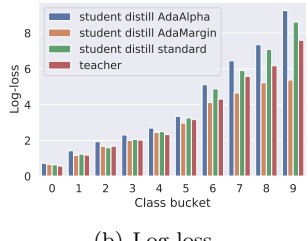
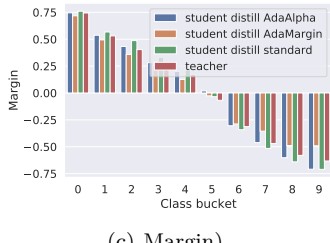

(a) Accuracy.  (b) Log-loss.  (c) Margin).

Figure 8: Logit statistics for ResNet-56 self-distillation on CIFAR-100 LT, for the teacher, self-distilled student, and our adaptive methods. Per Figure 4, we create 10 class buckets. AdaMargin flattens both the margin and log-loss distributions. AdaAlpha increases log loss across classes, while improving margins on few buckets, including flipping the bucket *5* to have positive margin.

Inspired by Dao et al. (2021), we consider a random forest based distillation setup: we use a teacher model that is a random forest *classifier* comprising 500 trees with a maximum depth of 20, and a student model that is a random forest *regressor* comprising 1 tree with a maximum depth of 20. The teacher model achieves a test (balanced) accuracy of 81.8%.

We perform distillation by feeding the student model the teacher's prediction scores, mixed in with the binary training labels with a weight $\alpha = 0.9$. Distillation improves the student's overall (balanced) accuracy significantly, from 76.2% to 79.3%. However, this gain is not distributed uniformly: using per-label subgroups, we find that distillation helps the positive class by $+7.4\%$, but hurts the negative class by $-1.2\%$. While by itself suggestive of asymmetry in distillation performance, the data admits an arguably more natural subgroup creation, based on available sex and sex features. For example, we find that amongst low-income males, distillation hurts by $-2.2\%$; further restricting to those who are Asian Pacific-Islander, the degradation is $-5.9\%$. This confirms that in scenarios where fairness may be a consideration, a naïve application of distillation may be inadmissible.

## C.4 Analysis of regularisation samples

Recently, Zhou et al. (2021) proposed the notion of *regularisation samples* to understand how distillation's performance can be improved. In brief, such samples correspond to cases where the teacher's prediction on the training label is less than the distilled student's prediction on this label; these may be shown to correspond to cases where a certain notion of "variance reduction" dominates a notion of "bias reduction". Given our analysis above of the asymmetric effects of distillation on certain subgroups, it is natural to consider whether or not these relate to the presence of regularisation samples in these groups.

Table 8: Difference between distillation and one-hot performance on Adult dataset. Here, subgroups are defined by the sex and label. $\Delta$ refers to the difference between the distilled and one-hot student's accuracy on the subgroup.

| sex | label | $\Delta$ |
|---|---|---|
| Male | 0 | -2.222 |
| Female | 0 | 0.393 |
| Female | 1 | 2.373 |
| Male | 1 | 8.384 |

Table 9: Difference between distillation and one-hot performance on Adult dataset. Here, subgroups are defined by the sex, race, and label. $\Delta$ refers to the difference between the distilled and one-hot student's accuracy on the subgroup.

| race | sex | label | $\Delta$ |
|---|---|---|---|
| Amer-Indian-Eskimo | Female | 1 | -66.667 |
| Asian-Pac-Islander | Male | 0 | -5.941 |
| Other | Male | 0 | -4.347 |
| Black | Female | 1 | -2.381 |
| White | Male | 0 | -2.248 |
| Black | Male | 0 | -1.639 |
| Black | Female | 0 | -0.140 |
| Asian-Pac-Islander | Female | 0 | 0.000 |
| White | Female | 0 | 0.388 |
| Other | Female | 0 | 2.439 |
| White | Female | 1 | 2.724 |
| Amer-Indian-Eskimo | Female | 0 | 6.349 |
| Amer-Indian-Eskimo | Male | 0 | 6.493 |
| Asian-Pac-Islander | Female | 1 | 7.692 |
| White | Male | 1 | 7.796 |
| Black | Male | 1 | 9.489 |
| Other | Male | 1 | 15.000 |
| Asian-Pac-Islander | Male | 1 | 19.626 |
| Other | Female | 1 | 20.000 |
| Amer-Indian-Eskimo | Male | 1 | 25.000 |

Figure 9(a) visualises the distribution of regularisation samples inside subgroups defined by 10 label buckets. where the labels are sorted in descending order of label frequency. Here, we compare the predicted probabilities of the teacher and final distilled student models on all *training* samples (as was done in the analysis of Zhou et al. (2021)). Interestingly, we see that the tail buckets tend to have very few regularisation samples; i.e., for rare labels, the teacher prediction on the training label is generally higher than that of the distilled student model. We confirm this in Figure 9(b).

While the analysis of Zhou et al. (2021) was primarily for training samples — since the aim in identifying regularisation samples was to mitigate their influence during training — we may also identify the breakdown of such samples on test data. Figure 10(a) shows that, compared to the training set, there are in absolute terms more such samples across nearly every label bucket; however, there is again no clear correlation between the label bucket and the fraction of such samples. In particular, the tail bucket is again the one with the *fewest* regularisation samples. This is corroborated by the probability scores of the teacher and student in Figure 10(b).

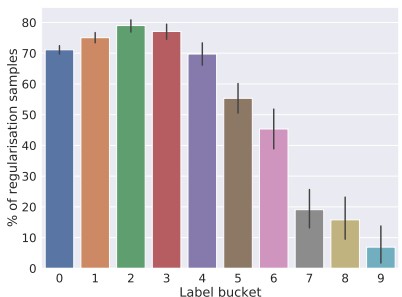

(a) Distribution of regularisation samples per label bucket.

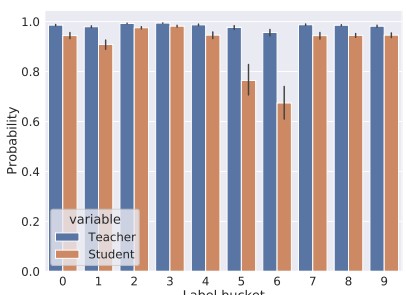

(b) Teacher versus distilled student probailities on training label.

Figure 9: Study of regularisation samples on training set, CIFAR-100 LT.

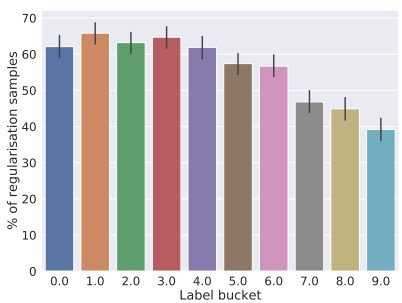

(a) Distribution of regularisation samples per label bucket.

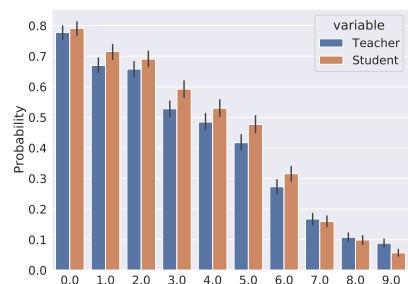

(b) Teacher versus distilled student probailities on test label.

Figure 10: Study of regularisation samples on test set, CIFAR-100 LT.

| Method | Per-class accuracy statistics | | | |
|---|---|---|---|---|
| | Mean | Worst 10 | Top 10% | $\Delta$10 |
| One-hot | 44.16 | 3.00 | 87.70 | 41.16 |
| Distillation 1$\times$ | 45.49 | 0.90 | 88.10 | 44.59 |
| Distillation 2$\times$ | 45.22 | 0.00 | 88.40 | 45.22 |
| Distillation 3$\times$ | 44.80 | 0.00 | 87.60 | 44.80 |

Table 10: Results of repeated distillation on CIFAR-100 LT. Using a distilled student as teacher for a subsequent round of distillation is seen to further hurt worst-class accuracy.

Overall, this results suggest that the existing notion of regularisation samples may not, by themselves, be sufficient to predict the poor performance of distillation on certain subgroups defined by labels.

### C.5 Impact of repeated distillation

In the body, we showed that performing distillation once can harm worst-class accuracy. However, what is the effect of repeating this process, and distilling using the resulting student as a new teacher? Does the worst-class accuracy get further harmed?

Table 10 shows that on CIFAR-100 LT, repeating distillation can indeed harm worst-class performance, even though average performance remains roughly similar. This further highlights the potential tension between average and worst-case performance under distillation.

Table 11: Ablations of design choices in the proposed methods: 1) remove distillation signal from the bottom 10% of classes, according to confidence; 2) randomly shuffle per-class $\alpha$ values; 3) weight distillation based on student and teacher confidence Zhou et al. (2021).

| Dataset | Method | Per-class accuracy statistics | | |
|---------|--------|------|---------|----------|
| | | Mean | Worst-1 | Worst-10 |
| CIFAR-100 | AdaAlpha | $75.52 \pm 0.10$ | $49.33 \pm 3.09$ | $56.59 \pm 0.44$ |
| | remove hardest 10% | $75.40 \pm 0.04$ | $48.33 \pm 1.89$ | $55.79 \pm 1.19$ |
| | shuffle temperatures | $74.56 \pm 0.93$ | $46.00 \pm 1.91$ | $53.10 \pm 1.22$ |
| | Zhou et al. (2021) | $75.74 \pm 0.20$ | $50.33 \pm 0.47$ | $54.93 \pm 0.34$ |

## C.6 Additional ablations

We confirm that the success of AdaAlpha is not immediately replicated by simpler baselines: (i) *remove hardest 10%*, which removes the distillation loss component on bottom 10% labels according to the per class margins found using Equation 4. It helps analyze whether there is any additional gain beyond simply removing teacher's supervision where it is arguably wrong. (ii) *shuffle temperatures*, which randomly shuffles the per-class $\alpha_y$ values used in AdaAlpha. This determines whether the precise choice of which labels to up- or down-weight is important; (iii) the adaptive distillation scheme of Zhou et al. (2021), where distillation is weighted differently across examples depending on the teacher and student scores.

In Table 11, we find that the first two methods work worse than the proposed AdaAlpha method, indicating that the precise choice of which labels to up- or down-weight is important, and that it does not suffice to merely ignore the teacher on entire subgroups.

## D Theoretical analysis

Consider the kernel ridge regression problem:

$$\min_{f_{\mathbf{w}} \in \mathcal{H}} \sum_{n=1}^{N} (f_{\mathbf{w}}(\mathbf{x}_n) - y_n)^2 + \lambda \|\mathbf{w}\|_2^2, \tag{7}$$

where $\mathcal{H} = \{f_{\mathbf{w}} \colon \mathbf{x} \mapsto \mathbf{w}^\top \Phi(\mathbf{x}) \mid \mathbf{w} \in \ell^2\}$. Let the *kernel* function $k(\mathbf{x}, \mathbf{x}') = \Phi(\mathbf{x})^\top \Phi(\mathbf{x}')$, and let $\mathbf{K} \in \mathbb{R}^{N \times N}$ be the kernel matrix with $\mathbf{K}_{ij} = k(\mathbf{x}_i, \mathbf{x}_j)$. Let $f^*$ denote the minimizer of equation 2, and $\mathbf{f}^*$ be the vector with $\mathbf{f}_n^* = f^*(\mathbf{x}_n)$. Then $\mathbf{f}^* = \mathbf{K}(\mathbf{K} + \lambda \cdot \mathbf{I}_N)^{-1}\mathbf{y}$, where $\mathbf{y}$ is the vector of targets with $\mathbf{y}_n = y_n$.

Now suppose $\mathbf{K}$ admits an eigendecomposition $\mathbf{V}^\top \mathbf{D} \mathbf{V}$, where $\mathbf{V}$ is an orthonormal and $\mathbf{D}$ a diagonal matrix. Consequently,

$$\mathbf{f}^* = \mathbf{V}^\top \mathbf{D} \left[ \mathbf{D} + \lambda \cdot \mathbf{I}_N \right]^{-1} \mathbf{V}\mathbf{y} = \mathbf{V}^\top \mathbf{A}\mathbf{V}\mathbf{y}, \tag{8}$$

where $\mathbf{A} = \mathbf{D} \left[ \mathbf{D} + \lambda \cdot \mathbf{I}_N \right]^{-1}$. From Mobahi et al. (2020), we know that using the predictions of $\mathbf{f}^*$ for distillation results in a solution

$$\mathbf{f}^{\text{dist}} = \mathbf{V}^\top \mathbf{A}^2 \mathbf{V}\mathbf{y}. \tag{9}$$

Recall that we consider the following data distribution for our analysis. Let $\mathbf{o}_i, i \in [L]$ be $L$ orthonormal vectors in $\mathbb{R}^D$ corresponding to the input representation $\mathbf{x}_i$ from different subgroups $L$. Let each subgroup have $d_i$ number of examples indicating the subgroup frequency. In this setting the kernel matrix ($\mathbf{K}_{ij} = \mathbf{x}_i^\top \mathbf{x}_j$) has a block diagonal structure, with blocks of ones size $d_i \times d_i$.

**Claim 3.** *Let $\mathbf{o}_i, i \in [L]$ be $L$ orthonormal vectors in $\mathbb{R}^D$. Let $d_1, \cdots, d_L$ be positive integers and $N = \sum_{i=1}^{L} d_i$. Let the dataset $\{\mathbf{x}_i\}_{i\in[N]}$ consist of $d_i$ samples of $v_i, \forall i$. Then the eigenvalues of the $N \times N$ kernel matrix $\mathbf{K}$ (where $\mathbf{K}_{ij} = \mathbf{x}_i^\top \mathbf{x}_j$), are $\{d_i\}_{i\in[L]}$.*

*Proof of Theorem 1.* Observe that the regression error is

$$
\begin{aligned}
\|\mathbf{f}^* - \mathbf{y}\|^2 &= \|\mathbf{V}^\top \mathbf{A} \mathbf{V} \mathbf{y} - \mathbf{y}\|^2 \\
&= \|\mathbf{V}^\top \mathbf{A} \mathbf{V} \mathbf{y} - \mathbf{V}^\top \mathbf{V} \mathbf{y}\|^2 + \|\mathbf{V}_\perp^\top \mathbf{V}_\perp \mathbf{y}\|^2 \\
&= \|\mathbf{V}^\top (\mathbf{A} - \mathbf{I}) \mathbf{V} \mathbf{y}\|^2 + \|\mathbf{V}_\perp^\top \mathbf{V}_\perp \mathbf{y}\|^2
\end{aligned}
$$

Here, $\mathbf{V}_\perp$ is the basis of the subspace orthogonal to $\mathbf{V}$ such that $\mathbf{V}^\top \mathbf{V} + \mathbf{V}_\perp^\top \mathbf{V}_\perp = \mathbf{I}$. Note that $\|\mathbf{V}_\perp^\top \mathbf{V}_\perp \mathbf{y}\|^2$ corresponds to the error in subspace orthogonal to the kernel. Hence, this error is same for all the models.

Let $\mathcal{M}_i$ denote the subset of data points from subgroup $i$. The mean error component in the subspace of examples from subgroup $i$ is

$$
\begin{aligned}
\frac{1}{d_i} \sum_{k \in \mathcal{M}_i} \|\mathbf{V}_k^\top (\mathbf{A} - \mathbf{I}) \mathbf{V} \mathbf{y}\|^2 &= \frac{1}{d_i} \sum_{k \in \mathcal{M}_i} \left( \frac{d_i}{d_i + \lambda} - 1 \right)^2 \mathbf{y}_k^2 \\
&= \left( \frac{\lambda}{d_i + \lambda} \right)^2 \bar{\mathbf{y}}_i^2.
\end{aligned}
$$

Here $\bar{\mathbf{y}}_i^2 = \frac{1}{d_i} \sum_{k \in \mathcal{M}_i} \mathbf{y}_k^2$ is the mean subgroup label. The above equations follow from noting that the eigenvectors of examples from subgroup $i$ ($\mathbf{V}_i$) are non-zero ($\frac{1}{\sqrt{d_i}}$) only for the indices corresponding to examples in subgroup and zero everywhere else.

Recall that the predictions after one round of self distillation are $\mathbf{f}^{\mathrm{dist}} = \mathbf{V}^\top \mathbf{A}^2 \mathbf{V} \mathbf{y}$. Hence, we can similarly compute the mean error of subgroup $i$ after distillation as follows:

$$
\begin{aligned}
\frac{1}{d_i} \sum_{k \in \mathcal{M}_i} \|\mathbf{V}_k^\top (\mathbf{A}^2 - \mathbf{I}) \mathbf{V} \mathbf{y}\|^2 &= \frac{1}{d_i} \sum_{k \in \mathcal{M}_i} \left( \frac{d_i^2}{(d_i + \lambda)^2} - 1 \right)^2 \mathbf{y}_k^2 \\
&= \left( \frac{\lambda^2 + 2\lambda d_i}{(d_i + \lambda)^2} \right)^2 \bar{\mathbf{y}}_i^2.
\end{aligned}
$$

Hence,

$$
\begin{aligned}
\epsilon_i - \epsilon_i^{\mathrm{dist}} &= \left( \frac{\lambda}{d_i + \lambda} \right)^2 \bar{\mathbf{y}}_i^2 - \left( \frac{\lambda^2 + 2\lambda d_i}{(d_i + \lambda)^2} \right)^2 \bar{\mathbf{y}}_i^2 \\
&= \left( \frac{\lambda^2 (d_i + \lambda)^2 - (\lambda^2 + 2\lambda d_i)^2}{(d_i + \lambda)^4} \right) \bar{\mathbf{y}}_i^2 \\
&= \left( \frac{\lambda^4 + \lambda^2 d_i^2 + 2\lambda^3 d_i - (\lambda^4 + 4\lambda^2 d_i^2 + 4\lambda^3 d_i)}{(d_i + \lambda)^4} \right) \bar{\mathbf{y}}_i^2 \\
&= \left( \frac{-3\lambda^2 d_i^2 - 2\lambda^3 d_i}{(d_i + \lambda)^4} \right) \bar{\mathbf{y}}_i^2.
\end{aligned}
$$

We can show that if $\lambda < \sqrt{d_i d_j}$, then

$$
\begin{aligned}
\epsilon_i - \epsilon_i^{\mathrm{dist}} &= \left( \frac{-3\lambda^2 d_i^2 - 2\lambda^3 d_i}{(d_i + \lambda)^4} \right) \\
&> \left( \frac{-3\lambda^2 d_j^2 - 2\lambda^3 d_j}{(d_j + \lambda)^4} \right) \\
&= \epsilon_j - \epsilon_j^{\mathrm{dist}}.
\end{aligned}
$$

$\square$

