# OpenReview forum: "Teacher’s pet: understanding and mitigating biases in distillation"
_TMLR — Accepted by TMLR_

### Review · Reviewer_fUxL · 2022-07-31

**Summary Of Contributions:**

This paper observes that distillation can harm performance on certain subgroups, e.g., classes with few associated samples, compared to the vanilla student trained using the one-hot labels. Some empirical studies are verified across many network architectures, datasets and class-imbalanced scenarios. This paper proposes to soften the teacher influence for subgroups where it is less reliable. Experiments on several image classification benchmarks show the effectiveness of the proposed method in subgroup performance.

**Requested Changes:**

1.Writing issue: in Line 3 of page 7, the definition of margin seems confused. Should the formulation of $\max_{y^{‘} \ne y}p^{‘}_{y}(x)$ be $\max_{y^{‘} \ne y}p_{y^{‘}}(x)$?
2.Some results of related logit-based KD methods [1,2,3] should be shown in Table 3, such as your referred works in page 9 (the section of Relation to existing work). Moreover, I also want to know whether the previous logit-based KD method would solve the subgroup performance degradation problem or not.
3.Some references of self-supervised KD methods [4,5] in third paragraph of page 3 are missing.
4.Open issue: beyond the image classification, could the proposed method apply to pixel-wise classification problem, for example, Hinton’KD in semantic segmentation [6,7]?

Reference:
[1] Rethinking soft labels for knowledge distillation: A bias–variance tradeoff perspective. ICLR2021.
[2] Prime-aware adaptive distillation. ECCV2020.
[3] Channel distillation: Channel-wise attention for knowledge distillation. CoRR, abs/2006.01683, 2020.
[4] Hierarchical Self-supervised Augmented Knowledge Distillation. IJCAI2021.
[5] Knowledge Distillation Using Hierarchical Self-Supervision Augmented Distribution. TNNLS-2022.
[6] Structured knowledge distillation for semantic segmentation. CVPR2019.
[7] Cross-Image Relational Knowledge Distillation for Semantic Segmentation. CVPR2021.


**Strengths And Weaknesses:**

Strengths:
1.	The proposed method is well-written and the motivation is clear and easy to understand.
2.	The problem is interesting and novel. Improving subgroup performance of KD seems a practical topic.
3.	The proposed method is simple but seems effective in subgroup performance. The theoretical proof looks reasonable.
4.	The preliminary experiments make sense to su
5.	pport the motivation.

Weakness:
1.	The experiments are not sufficient. This paper does not compare some related works for logit-based KD methods. And the experimental benchmark is only image classification task but does not transfer to other recognition tasks.
2.	The improved performance using the proposed method on overall accuracy seems very marginal, which may limit the usefulness of the proposed method in practice.

---

> ### Author Response · Authors · 2022-08-22
> **Response to Reviewer fUxL**
>
> Thanks for the review! Please find our responses below. We updated the draft with the suggested changes, highlighting the changes in blue.
>
> At the outset, we would like to emphasize that a bulk of the paper comprises an analysis of *when* and *why* distillation harms subgroups. As noted in the Introduction section after listing the contributions, we view this as our primary contribution. The bulk of the paper is on analysis of the *causes* of non-uniform gains in distillation in Section 3, whereas we only spend a single section discussing new methods (Section 4).
>
> > *Requested change 1. Writing issue: in Line 3 of page 7, the definition of margin seems confused. Should the formulation of $\max_{y^{‘} \ne y}p^{‘}{y}(x) be \max{y^{‘} \ne y}p_{y^{‘}}(x)$?*
>
> We updated the typo, thanks for catching it!
>
> > *Requested change 2. Some results of related logit-based KD methods [1,2,3] should be shown in Table 3, such as your referred works in page 9 (the section of Relation to existing work). Moreover, I also want to know whether the previous logit-based KD method would solve the subgroup performance degradation problem or not.*
>
> Thanks for this suggestion. To address it, we added Table 5 in the updated manuscript, where we compare our methods against the method from Zhou et al. (2021). We find
> that Zhou et al. (2021) is competitive in terms the mean accuracy, but, contrary to the Ada*
> methods, does not mitigate the issue of distillation harming accuracy over the worst classes.
>
> > *Requested change 3. Some references of self-supervised KD methods [4,5] in third paragraph of page 3 are missing.*
>
> We added the references to self-supervised KD methods suggested [4,5], thanks!
>
> > *Requested change 4. Open issue: beyond the image classification, could the proposed method apply to pixel-wise classification problem, for example, Hinton’KD in semantic segmentation [6,7]?*
>
>
> We agree that the question of whether distillation can disproportionately harm hardest subgroups beyond the classification scenario is very valuable. However, we note even the classification setup hasn’t been studied in this regard before. Here, we focused on classification, as even within the classification setup we explored a wide range of scenarios: varying the number of classes, distribution of classes, architectures for the teacher and the student… This made the work already quite dense. Again, we agree that including a wider range of setups will be valuable for future work.
>
> > *Weakness 1. The experiments are not sufficient. This paper does not compare some related works for logit-based KD methods. And the experimental benchmark is only image classification task but does not transfer to other recognition tasks.*
>
> Per reviewer’s comment, we compare Ada* methods against the Zhou et al. method to Table 5. As can be seen, Zhou et al. does not fix the subgroup performance issues.
>
> While we indeed mostly considered classification tasks on image datasets, we note that this practice is in line with a series of other works that seek to better understand deep learning:
>
> Zhang et al., “Understanding deep learning requires rethinking generalization”, ICLR 2017.
> Muller et al., “When Does Label Smoothing Help?”, NeurIPS 2019.
> Neyshabur et al., “What is being transferred in transfer learning?”, NeurIPS 2020.
> Jiang et al., “Characterizing Structural Regularities of Labeled Data in Overparameterized Models”, ICML 2021.
> Zhou et al., “Rethinking Soft Labels for Knowledge Distillation: A Bias-Variance Tradeoff Perspective”, ICLR 2021
>
> Please note that beyond image classification datasets, we also considered the UCI Adult dataset. We agree that expanding the number of approaches would be of interest. Here, we focus on conducting extensive experiments on varying: dataset size, architectures, label skew.
>
> > *Weakness 2. The improved performance using the proposed method on overall accuracy seems very marginal, which may limit the usefulness of the proposed method in practice.*
>
> The methods indeed exhibit less impact on the average accuracy than the worst-class accuracy. However, we do manage to show improvements in all considered datasets on worst-class accuracy (see Table 3).

---

### Review · Reviewer_PJm1 · 2022-08-07

**Summary Of Contributions:**

In the field of knowledge distillation, the issue of error amplification on subgroups with few samples is a problem that has ample space for exploration and has great practical value in both academia and industry. This issue is well defined for the first time in this paper. Moreover, it analyzes and explains this issue through theoretical and empirical lenses. This paper has also proposed two approaches to correct the traditional distillation method, which are both simple but effective. Nevertheless, the experiment is incomplete and therefore insufficient to prove the superiority of these two methods. To my knowledge, the most commonly used methods to solve the class imbalance problem are data augmentation, ensemble learning, etc. At least, some tricks may also help, for example, only distill on the subgroups that the teacher predicts more correctly. The comparison with stronger baselines can better prove the practical value of the proposed method.


**Requested Changes:**

- Take a complete literature review and categorize the popular methods. e.g.:
  - Sampling-based: Class-Balanced Distillation for Long-Tailed Visual Recognition
  - Reinforcement-learning-based: Reinforced knowledge distillation: Multi-class imbalanced classifier based on policy gradient reinforcement learning
  - Data-augmentation-based: Solving the Class Imbalance Problem Using a Counterfactual Method for Data Augmentation
  - Ensemble-learning-based: A Review on Ensembles for the Class Imbalance Problem: Bagging-, Boosting-, and Hybrid-Based Approaches
- Choose some well-known ones as additional baselines such as data augmentation, ensemble learning, or more tricks.
- Give more theoretical analysis and prove that the existing methods do not work theoretically (and experimentally).

**Strengths And Weaknesses:**

## Pros :
- The problem is well defined and crucial.
- The pilot experiment is complete and rigorous, giving a variety of ablations with different variables.
- The theoretical analysis is correct and strictly derived.
- The proposed methods are simple but effective.

## Cons :
- Literature survey is not sufficient, which lacks some necessary approaches and therefore cannot summarize the background in a complete way.
- The baselines are not strong enough. Some traditional or recent methods for overcoming the class imbalance issue are not presented.

---

> ### Author Response · Authors · 2022-08-22
> **Response to reviewer PJm1**
>
> > *Take a complete literature review and categorize the popular methods. e.g.:
> Sampling-based: Class-Balanced Distillation for Long-Tailed Visual Recognition
> Reinforcement-learning-based: Reinforced knowledge distillation: Multi-class imbalanced classifier based on policy gradient reinforcement learning
> Data-augmentation-based: Solving the Class Imbalance Problem Using a Counterfactual Method for Data Augmentation
> Ensemble-learning-based: A Review on Ensembles for the Class Imbalance Problem: Bagging-, Boosting-, and Hybrid-Based Approaches*
>
> Thanks for the suggestion. We have updated **Section 4.2** to include a more detailed overview of the long-tailed learning literature.
>
> As noted in the updated text, however, we would like to emphasize two points. First, *the primary contribution of this work is in systematically identifying the causes for subgroup performance degradation under distillation*. Second, this subgroup degradation can occur *even when classes (or subgroups) are perfectly balanced*. Thus, there is no direct translation of such techniques to the distillation setting considered in this paper. We certainly agree that further exploring extensions of existing methods to this setting is of interest (e.g., perhaps one can modify sampling-based methods to account for subgroup hardness); however, we think this would be worthy of a separate work in itself.
>
> > *Choose some well-known ones as additional baselines such as data augmentation, ensemble learning, or more tricks.*
>
> Thanks for the suggestion. We agree that these are very interesting ideas. We would like to emphasize that it is not a priori clear how techniques for long-tail learning should be adjusted for tackling the problem of improving the worst **subgroup** performance, where subgroups **may be perfectly balanced** (e.g., our results on CIFAR-100 and ImageNet). Studying such an extension would certainly be interesting, but we think more suitable for a separate work in itself.
>
> Further, we would like to re-iterate that a bulk of the paper comprises an analysis of *when* and *why* distillation harms subgroups. As noted in the Introduction section after listing the contributions, we view this as our primary contribution. The bulk of the paper is on analysis of the *causes* of non-uniform gains in distillation in Section 3, whereas we only spend a single section discussing new methods (Section 4).
>
>
> > *Give more theoretical analysis and prove that the existing methods do not work theoretically (and experimentally).*
>
> Thanks for the suggestion. While we completely agree that further theoretical analysis would be useful, we hope that the initial analysis we have provided (on the potential of distillation to harm under a particular data distribution; and the Bayes-consistency of the Ada-Margin method) can serve as a useful starting point for future work. We note also that theoretical analysis of regular distillation (aside from the issue of worst-subgroup performance) is itself developing, and so a complete analysis of the effect of distillation on worst-subgroup error would be a sizable effort that could be a separate work in itself.

---

### Review · Reviewer_UU7c · 2022-08-08

**Summary Of Contributions:**

This work systematically studies a previously under-explored issue with knowledge distillation: though distillation usually improves the overall accuracy of the student model, the accuracy on some rare subgroups may be deteriorated by distillation. Empirical results and theoretical analysis help to identify that a large number of classes and a skewed class distribution can contribute to this observed phenomenon. Two solutions, AdaAlpha and AdaMargin, are proposed to mitigate this unbalanced subgroup improvement issue in knowledge distillation, by limiting the teacher influence for the subgroups where the teacher is making confident but incorrect predictions.

**Broader Impact Concerns:**

No concerns.

**Requested Changes:**

1)	As pointed out in the Weaknesses, the major concerns are with the proposed two solutions. More evaluation on different distillation settings, and explanation about the relative improvements over the baselines would be appreciated.

2)	Section 3: The empirical study of the bias in distillation is based on a fixed value of alpha (a weight for balancing the ground-truth labels and teacher predictions), set to 1.0 for CIFAR-100 and 0.2 for ImageNet. In fact, this weight parameter leads to different behaviors regarding the teacher model’s influence. It would be great to show how the distillation bias between subgroups changes with alpha.

3)	Figure 1: Before reading Section 3, a reader may have the following questions about how the “worst subgroups” are selected: 1) Are they the subgroups with the worst performance improvement? If so, why is the performance improvement for the worst subgroup in Figure 1(b) - Adaptive Margin larger than average? 2) How are the “worst subgroups” related to the “rarest classes” described in the caption? The frequency of samples may not be always consistent with performance or performance improvement in an unbalanced dataset. A bit more explanation about the “worst subgroups” in the caption could be helpful.

4)	Section 3.2: EfficientNets (Tan & Le, 2021) are a family of architectures with varying performance and model sizes. It would be better to point out the specific architecture (B0-B7) used in the experiments.


**Strengths And Weaknesses:**

Strengths:

1)	This work identifies a previously underexplored issue with knowledge distillation. Distillation is a widely applied technique for improving model accuracy while preserving the model size. This systematic study raises the awareness of fairness between subgroups in a distillation setting, and helps future research to improve the fairness when using the distillation technique in various applications.

2)	This work has conducted a thorough evaluation of the subgroup bias issue in multiple settings and benchmarks. This study demonstrates that a) this bias issue in distillation can happen in various datasets and model architectures, and b) distillation can be biased due to the large number of classes, the class imbalance, and other factors related to fairness.

3)	In addition to empirical results, this work also takes a theoretical perspective to understand the subgroup bias in distillation, based on the analysis of a ridge regression problem.

4)	The writing is clear and easy to follow.

Weaknesses:

The major weaknesses are related to the proposed two solutions, AdaAlpha and AdaMargin.

1)	Limited improvement: From Table 3, the relative improvement from standard distillation seems insignificant. For example, for the worst class in ImageNet, the standard distillation reaches 14.00±1.41 accuracy, and the proposed two solutions can only improve that to 15.50±2.18 and 15.00±1.00. For the long-tailed (LT) versions of CIFAR-100 and ImageNet, the accuracy of the worst classes is still close to zero. In addition, AdaMargin may not converge in some cases.

2)	Both solutions require a holdout split to quantify the teacher model’s influence for different subgroups, which may be an undesired necessity: a) Sampling the holdout split introduces additional variance and complexity. The authors also consider that the divergence of AdaMargin may be related to the holdout split. b) In a long-tailed or few-shot learning setting where some classes only contain a handful of examples, splitting out a holdout set may be unaffordable and hurt final performance.

3)	The solutions are only examined in the self-distillation setting. It is unclear whether they are helpful when the teacher and student architectures are different.

---

> ### Author Response · Authors · 2022-08-22
> **Response to Reviewer UU7c**
>
> > *Both solutions require a holdout split to quantify the teacher model’s influence for different subgroups, which may be an undesired necessity: a) Sampling the holdout split introduces additional variance and complexity. The authors also consider that the divergence of AdaMargin may be related to the holdout split. b) In a long-tailed or few-shot learning setting where some classes only contain a handful of examples, splitting out a holdout set may be unaffordable and hurt final performance.*
>
> We agree that this is a limitation of our method. Please note that many techniques in the long tail and fairness literature require access to a hold out split or estimate certain quantities using cross validation, e.g.:
>
> Dao et al. 2021, “Knowledge Distillation as Semiparametric Inference”, ICLR 2021
>
> Patrini et al. Making Deep Neural Networks Robust to Label Noise: a Loss Correction Approach, CVPR 2017
>
> Menon et al. 2021, Long-tail learning via logit adjustment, ICLR 2021
>
>
>
> > *Section 3: The empirical study of the bias in distillation is based on a fixed value of alpha (a weight for balancing the ground-truth labels and teacher predictions), set to 1.0 for CIFAR-100 and 0.2 for ImageNet. In fact, this weight parameter leads to different behaviors regarding the teacher model’s influence. It would be great to show how the distillation bias between subgroups changes with alpha.*
>
> Thanks for the suggestion. To address your suggestion, we ran additional experiments on CIFAR-100 LT where we swept the mixing weight hyper-parameter and added the results in the new Figure 3. We find that across architectures, for all choices for the mixing parameter alpha, distillation improves the average accuracy, but harms the worst-10 class accuracy.
>
> > *The solutions are only examined in the self-distillation setting. It is unclear whether they are helpful when the teacher and student architectures are different. As pointed out in the Weaknesses, the major concerns are with the proposed two solutions. More evaluation on different distillation settings, and explanation about the relative improvements over the baselines would be appreciated.*
>
> Thanks for this suggestion. To address it, we added the new Table 4 into the write-up, where we consider different teacher-student architecture pairs. We find that our proposed methods improve worst class performance across considered scenarios.
>
> > *Figure 1: Before reading Section 3, a reader may have the following questions about how the “worst subgroups” are selected: 1) Are they the subgroups with the worst performance improvement? If so, why is the performance improvement for the worst subgroup in Figure 1(b) - Adaptive Margin larger than average? 2) How are the “worst subgroups” related to the “rarest classes” described in the caption? The frequency of samples may not be always consistent with performance or performance improvement in an unbalanced dataset. A bit more explanation about the “worst subgroups” in the caption could be helpful.*
>
>
> Thanks for these great suggestions. We added the suggested clarification in the Figure 1 caption. We also checked how the ordering according to frequencies and according to the teacher's performance relate on the unbalanced set. We find them to be strongly related: for CIFAR-100 LT, the Spearman rank correlation coefficient is equal to 0.84.
>
> > *Section 3.2: EfficientNets (Tan & Le, 2021) are a family of architectures with varying performance and model sizes. It would be better to point out the specific architecture (B0-B7) used in the experiments.*
>
> We use Efficient-NetV2 L following the work from Tan and Le 2021. We clarified in the updated draft.

---

> > ### Comment · Reviewer_UU7c · 2022-09-06
> > **Review Response**
> >
> > Thanks for your response! In the updated draft, new experiments with various distillation hyper-parameters and teacher-student architectures successfully demonstrate that subgroup bias is common in distillation, and the proposed AdaAlpha/AdaMargin methods can balance performance improvements in more scenarios. My previous concerns about them are addressed by this response.
> >
> > However, my concerns about the practicability of the two proposed methods still remain: 1) The relative improvement compared to baselines is marginal. 2) A held-out split is indispensable, which is undesired if some subgroups only have highly limited samples. I do not have further concerns other than the proposed methods. Overall, the main analysis of distillation hurting some subgroups is well established in this work.

---

### Review · Reviewer_zieG · 2022-08-09

**Summary Of Contributions:**

This paper considers the effectiveness of Knowledge Distillation (KD), especially for the case where the label distribution is highly non-uniform. From experiments, the authors find that KD can hurt subgroup performance since, under a long-tail label distribution, the teacher might not have the correct information for subgroup data. To resolve the subgroup performance issue, the authors propose two adaptive methods. The first one introduces adaptive mixing weights that control the weight in front of the KD loss between the teacher and the student so that well-trained classes have high weights while incorrectly trained classes have low weights. The second approach uses a per-class margin loss which enforces minor classes having more margin. The experimental results show the two proposed approaches outperform the vanilla KD.

**Requested Changes:**

1. More experiment results with various different hyper-parameters and teacher and student architectures are needed.

2. More discussions on mixing weight and adaptive margin functions might be necessary.

3. In Section 3.3, when we sub-sample the classes, the number of data points is also changed. I think the number of data points is a very important factor for the KD performance. So, the authors require to check the performance when the numbers of data points are the same, but the numbers of classes are different.

**Strengths And Weaknesses:**

Strength

1. This paper provides a new perspective on KD about fairness and minority.   The new direction seems important.

2. The authors find an interesting observation KD can hurt minorities and tries to explain the reasons for the observation with experiments and a theorem.

3. This paper is well-written and easy to follow.

Weakness

1. The experiment part needs to be improved. KD has some important hyper-parameters like the temperature and the mixing weight. The authors should provide more results with various settings and show some trends. Moreover, NN architectures of the teacher and the student are very important factors for the performance. The authors should consider various scenarios.

2. The two proposed adaptive distillation methods are interesting ideas. However, I think we could think of many different mixing weight functions and per-class margin functions that function almost the same way. It would be very nice if the authors could provide more discussions on that.

---

> ### Author Response · Authors · 2022-08-22
> **Response to reviewer zieG**
>
> > *The experiment part needs to be improved. KD has some important hyper-parameters like the temperature and the mixing weight. The authors should provide more results with various settings and show some trends. Moreover, NN architectures of the teacher and the student are very important factors for the performance. The authors should consider various scenarios. More experiment results with various different hyper-parameters and teacher and student architectures are needed.*
>
> Thanks for the suggestion. To address your suggestion, we ran additional experiments on CIFAR-100 LT where we swept the temperature, the mixing weight hyper-parameter and the model architecture, and added the results in the new Figure 3. We find that across architectures, for all temperatures and choices for the mixing parameter alpha, distillation improves the average accuracy, but harms the worst-10 class accuracy.
>
> Please note we also included sweeping of student and teacher architectures over the ImageNet dataset in Table 2, where we find that distillation consistently harms worst class accuracy across teacher and student architectures.
>
>
> > *The two proposed adaptive distillation methods are interesting ideas. However, I think we could think of many different mixing weight functions and per-class margin functions that function almost the same way. It would be very nice if the authors could provide more discussions on that. More discussions on mixing weight and adaptive margin functions might be necessary.*
>
> We agree that there are other possible choices for the mixing strategies beyond AdaAlpha and AdaMargin methods we proposed. The ones we consider are simple and, as demonstrated in Table 3, prove effective. Moreover, the AdaMargin approach is a natural extension of existing ideas from long-tail learning and from that we obtain a consistency guarantee (see Lemma 2).
>
> > *In Section 3.3, when we sub-sample the classes, the number of data points is also changed. I think the number of data points is a very important factor for the KD performance. So, the authors require to check the performance when the numbers of data points are the same, but the numbers of classes are different.*
>
> Thanks for the suggestion. To address your comment we contrast results from experiments on the pairs of datasets (CIFAR-10, CIFAR-100), and (CIFAR-10 LT, CIFAR-100 LT). In each of these pairs of datasets, the set of the train examples is fixed but the number of classes changes from 10 to 100.
>
>
> We have added these results in Table 1. We find that the accuracy over the worst 10% classes according to the teacher does not drop for neither CIFAR-10 nor CIFAR-10 LT upon distillation. This supports the observation that limiting the number of classes thwarts the phenomenon of distillation harming the worst classes.

---

> > ### Comment · Reviewer_zieG · 2022-08-29
> > **Thank you for the answers**
> >
> > The authors have addressed all my concerns.

---

### Decision · Action_Editors · 2022-09-23

**Recommendation:** Accept with minor revision

**Comment:**

This paper contributes two interesting methods for mitigating KD biases. All the reviewers acknowledge the novelty and clarity of the papers. One of the reviewers questioned the literature review and experiment completeness and the authors partially addressed them, maybe that's the reason that the reviewer is leaning towards rejection. AE read the paper, reviews, authors' responses, and the revision, and indeed feels that the authors may want to further strengthen the survey part and the motivation of the experimental design. Overall, the merits of the paper clearly outweigh the demerits. AE recommends accept with further minor changes.

---

> ### Author Response · Authors · 2022-10-18
> **Thanks for the decision regarding our paper.**
>
> Thanks for the decision regarding our paper.
>
> We addressed the requested changes as follows: we added more discussion in Section 4 regarding the motivation of the experimental design, and we strengthened the survey part in Section 4. (This is an expanded version of a survey we included in response to the initial reviews.)
>
> We highlighted the changes in the revised version. Please let us know if any further changes are needed, or if we should submit the camera ready version of the paper.